# BEERL: Both Ends Explanations for Reinforcement Learning

**Ahmad Terra** [1,2,*], **Rafia Inam** [1,2] and **Elena Fersman** [2,3]

1 Ericsson AB, Royal Institute of Technology, 164 83 Stockholm, Sweden
2 Machine Design, Industrial Engineering and Management , KTH Royal Institute of Technology, 114 28 Stockholm, Sweden
3 Ericsson Inc., Royal Institute of Technology, Santa Clara, CA 95054, USA
* Correspondence: ahmad.terra@ericsson.com or terra@kth.se

**Abstract:** Deep Reinforcement Learning (RL) is a black-box method and is hard to understand because the agent employs a neural network (NN). To explain the behavior and decisions made by the agent, different eXplainable RL (XRL) methods are developed; for example, feature importance methods are applied to analyze the contribution of the input side of the model, and reward decomposition methods are applied to explain the components of the output end of the RL model. In this study, we present a novel method to connect explanations from both input and output ends of a black-box model, which results in fine-grained explanations. Our method exposes the reward prioritization to the user, which in turn generates two different levels of explanation and allows RL agent reconfigurations when unwanted behaviors are observed. The method further summarizes the detailed explanations into a focus value that takes into account all reward components and quantifies the fulfillment of the explanation of desired properties. We evaluated our method by applying it to a remote electrical telecom-antenna-tilt use case and two openAI gym environments: lunar lander and cartpole. The results demonstrated fine-grained explanations by detailing input features' contributions to certain rewards and revealed biases of the reward components, which are then addressed by adjusting the reward's weights.

**Keywords:** explainability; deep reinforcement learning; explainable reinforcement learning; reward decomposition; reward prioritization; bias

## 1. Introduction

Reinforcement learning (RL) is a method by which an autonomous agent learns from its interaction with the environment. This agent gathers information about its environment via state variables as input features and, based on it, calculates the quality of the current state to select an action to be generated as the output. In each step, a reward is given in return to measure how good the state is, especially in achieving the main goal of the task. One of the RL methods is deep reinforcement learning (DRL), and it has been used to solve various tasks with outstanding performance [1,2]. The main driving factor for high-performance DRL is the use of a deep neural network (DNN). In general, DNN is used in many complex tasks, such as identifying images, recognizing voices, generating fake videos, etc. Each node in DNN is a mathematical operation constructed in a certain way, and following all operations to generate the output is not intuitive. Since the process of how DNN produces a specific output is hard to understand and explain, explainable reinforcement learning (XRL) methods produce information about the AI model's behavior. Attributive explanation is the most common type of explanation [3], where the importance of an input features indicates how it affects the output of the model without informing the inner part of the model. Alternatively, a contrastive explanation shows why one action is preferred over the other, which may require model decompositions for RL the system. When the output of an RL model is decomposed into several components, a model may prefer one component over the other; we refer to this as component bias. Please note that, the bias of

the reward component is derived from the base value of [4] where it previously showed the bias of each class/action. However, the correlation of the input features importance with reward components has not been found in the literature. Additionally, the effect of reward prioritization is also not explored with respect to generating explanations. We present a novel "Both Ends Explanations for RL (BEERL)" method that produces correlations between input features and output reward components. The goals of our work are to generate more detailed explanations, to show the correlations among them, to summarize extensive information for the user, and to identify and mitigate the bias of the RL model.

The main contribution of BEERL method is to generate fine-grained explanations (a fine-grained explanation consists of input features and reward components pairs instead of either only input features or only reward components) by performing the following: (1) Exposing the effects of reward prioritization resulting in two different levels of the RL agent (Q-function and RL level); (2) Identifying the bias of the reward components and demonstrating a method to mitigate it; (3) Extending the contrastive explanation method to highlight a certain input feature(s) or reward component(s); (4) Formulating a focus value that provides a quantification of the explanation by measuring how much the desired behavior is satisfied. We apply our method to a remote electrical antenna-tilt use case and two openAI gym environments to evaluate the results. The result showed that the generated explanations can be used to evaluate the performance of the components of the RL agent. With these explanations, we also demonstrate several possible options to reconfigure the RL agent, including a bias reduction in the reward components.

**Paper outline:** Section 2 presents the background and related works. Section 3 describes the details of our BEERL method. Section 4 presents the application of BEERL to the use case and the experimental result. Section 5 summarizes the paper and points to possible extensions of future work. Appendix A presents the improved MSX formulation that is used to compress contrastive explanation. Appendixes B and C present the parameters of the RL agent and antenna tilt configuration. Appendix D shows the implementation of BEERL in two openAI gym environments.

## 2. Background and Related Works

This section elaborates the motivation of our work, the problem we are addressing in the current XRL method, and the relationship with other explainable artificial intelligence (XAI) methods.

### 2.1. Background

In XAI, generating an explanation can be performed intrinsically if the model is transparent (such as decision tree and a rule-based system) or after the AI model has produced its output (post hoc methods). Several post hoc methods such as SHAP [4] and LIME [5] are model-agnostic, which means that they are applicable to any black-box model such as neural networks (NN), ensemble models, etc. One type of XAI methods focuses on feature importance where knowing the contribution of different input variables is important to understand the characteristics of certain input variables. It may also indicate which factors/variables are not important and can be changed without affecting the output or which ones are important and must be retained to keep the same output. In the RL problem, this information is closely related to the input data used, specifically the states of the environment during the training phase. Deeplift [6], SHAP [4], and LIME [5] are methods that can be used to explain the feature importance of the DRL agent with respect to the actions. However, a method to measure the contribution of input features to the components of the RL reward function is lacking in the literature.

In RL, the reward function is designed to quantify the desired situation as feedback to update the agent's parameters. Designing and obtaining feedback on the formulation of the reward function is challenging. Its calculation generally considers several factors, but without a reward decomposition method, such as Hybrid Reward Architecture (HRA) [7], only a single value is given to the agent. HRA uses several reward values (respective of the

components) to update the RL agent's parameters. The HRA authors mentioned that the reward component typically depends on a subset of features without explaining this dependency further. Juozapaitis et al. [8] used HRA to explain the output layer of a DRL agent, specifically the RL reward components. However, it only presents the values of reward components without showing the bias and effect of different weights for each component.

A correlation of input and output explanations is lacking in the literature. With the available methods, the contributions of the input features are not reflected in the explanation of the reward component. Conversely, the feature importance does not inform us about the reward component to which the state variable contributes. In this work, we present a method for expanding these techniques not only to correlate input and output explanations but also to expose the effect of the reward weight and to identify and mitigate the bias on different reward components.

### 2.2. Related Works

This section presents related work on methods that focus on measuring the importance of the input feature to the output of the model. Then, it discusses the reward decomposition to complement the generated explanation in the output or the final layer.

### 2.2.1. Explainability for Input Features

Several XAI methods [6,9–12] have been developed for a specific type of AI model. Methods that explain the DNN model's [6,11,12] operation by tapping or backpropagating the inference process, resulting in faster computations than methods with repetitive model inferences. In order to understand a Deep Q-network (DQN), Zahavy et al. [13] introduced a method that requires manual clustering to generate an explanation in the form of a saliency map. On the other hand, Atrey et al. [14] manipulated the RL environment and used the saliency map to verify the counterfactual explanation and showed that the saliency explanation is subjective and insufficient. In that work, several attributive XAI methods were analyzed, but SHAP [4] was not included. Wang et al. [15] highlighted the importance of the baseline data that are required for the SHAP method in the RL problem but did not provide a detailed data selection method nor its verification. For a machine learning problem, SHAP [4] is a model-agnostic method in which it is observed to produce the best results when applied to the telecommunications use case [16]. RL-SHAP [17] is an example where SHAP [4] can be applied to the RL problem to explain the contribution of the actor's input features. A recent survey by Hickling et al. [18] also identified that SHAP is highly adopted in RL, especially in the field of vehicle guidance, robotic manipulation, and system control. In the aforementioned methods, they used the entire model to generate the explanation, while our method splits the model into each reward component, and the importance of the input feature is now generated per reward component, thus providing fine-grained explanations.

### 2.2.2. Explainability for Output/Reward Components

When a reward function considers multiple factors, the reward value can be decomposed by exposing its calculation [7]. Lin et al. decomposed the reward while disentangling the representation without domain knowledge [19]. These methods did not consider the contribution of every input feature, while our work connects it to the reward components. Juozapaitis et al. implemented reward decomposition and used the difference between Q-values of the contrasted actions to explain the agent's decision [8]. They introduced the concept of minimal sufficient explanation (MSX) to compress the contrastive explanation. Lin et al. in [20] used this concept to generate contrastive explanations by implementing an embedded self-prediction model via general value functions (GVFs) [21] that are located between two NNs used by the agent. In the above methods, all components are treated as the same (having equal weights, i.e., 1) and only final reward components values are obtained. Our method decouples the reward weights and Q-functions approximators, allowing each component to be treated equally and shows the weights explicitly. It uses the contributions

of the input features to each reward component to explain why an action is preferred over the other. Since our method generates a more granular explanation than earlier methods, we also propose a method to compress it. Another work by Bica et al. proposed a method to learn the reward weight using the counterfactual inverse RL algorithm [22] while our method maintains a static weight that encourages transparency during training.

### 2.2.3. Quantification of the Explanations

In XAI, the quality of the explanation is measured on a case-by-case basis because the explanations can be presented in various forms. Mohseni et al. in [23] summarized the evaluation measures for XAI, comprising different aspects such as the mental model, usefulness and satisfaction, user trust and reliance, human-AI task performance, and computational measures. Schmidt and Biessmann [24] presented the quality and trust metrics for AI explanation that are based on response time, mutual information, and agreement between XAI and humans. Zhang et al. [25] used KL-divergence to measure suboptimal attributes in an explanation to detect bias and failure. Anderson et al. [26] performed a more similar evaluation to our work, where they evaluated the improvement of the human mental model after being presented with a certain type of explanation. In that work, both input and output explanations were presented without any correlation. Our proposed work presents not only the correlation between input and output explanations but also a focus value to quantify the desired properties set by the user.

### 2.2.4. Other XRL Methods

In addition to the above methods, there are several other XRL methods which use the causal model [27], and distill the trained agent into a transparent model [28,29]. Madumal et al. [27] proposed a method that uses a causal model to explain the cause and counterfactual explanation of the behavior of the RL agent. This method requires that the causal model be prepared manually, which can inhibit implementation, especially when an expert is not available to build or verify it. Verma et al. [28] proposed an RL method in which they initially trained a DRL agent and derived transparent programmatic policies. Liu et al. [29] approximated a DRL Q-function using linear model U-trees where the leaf node of the tree is a linear model. Our method is different from the above because, instead of distilling the RL policy into a transparent model, it exposes the reward components and their proportion to make the second-last layer of the NN transparent, producing explanations at a finer level of granularity. With the existing methods [4–6,8,11,12], the quantity of the generated explanation will be the sum of the number of input features ($N_I$) and the number of reward components ($N_C$), while our method can produce $N_I \times N_C$ explanations.

### 2.2.5. Explainability of Remote Electrical Antenna Tilt

We apply our method to the remote electrical antenna tilt use case, which has previously been explored from different angles using state-of-the-art methods. Vanella et al. [30] applied a safe RL method to avoid the risk of performance degradation, especially in the exploration phase. In [31], Vanella et al. also introduced the offline RL method to learn the agent's optimal policy from the real-world data. Furthermore, Vanella et al. [32] used a contextual linear bandit method to find an optimal policy with fewer data than naive or rule-based algorithms. Bouton et al. [33] introduced an RL method that uses a predefined coordination graph to allow coordination among agents in controlling the antennas. Lastly, Jin et al. [34] proposed graph-attention Q-learning where the graph-attention network [35] is combined with DQN [1] to capture broader network information without having a large state for the agent. All these previous works do not focus on the explainability of the agent. Our work addresses this matter, including how to detect and mitigate bias. We implement our method and analyze our results on this use case and it can be implemented in any other environment, such as for OpenAI gyms (Cartpole and Lunar lander examples are shown in Appendix D).

## 3. Both End Explanations for Reinforcement Learning (BEERL) Method

The system designer determines the RL system to be used based on the task and the environment of the use case. In case the task provided by the system designer is of a high complexity, the produced machine learning model will be of a high complexity as well, and its understanding will be challenging for a human being. One of the main goals of explainability is to improve the human understanding of AI [36]. Therefore, we categorize the three main aspects of XRL: the environment in which the RL agent operates, the human user of the system, and the RL framework that performs its operation. The Both End Explanations for Reinforcement Learning (BEERL) method and the interplay between the XRL aspects are presented in Figure 1 and discussed in the following subsections.

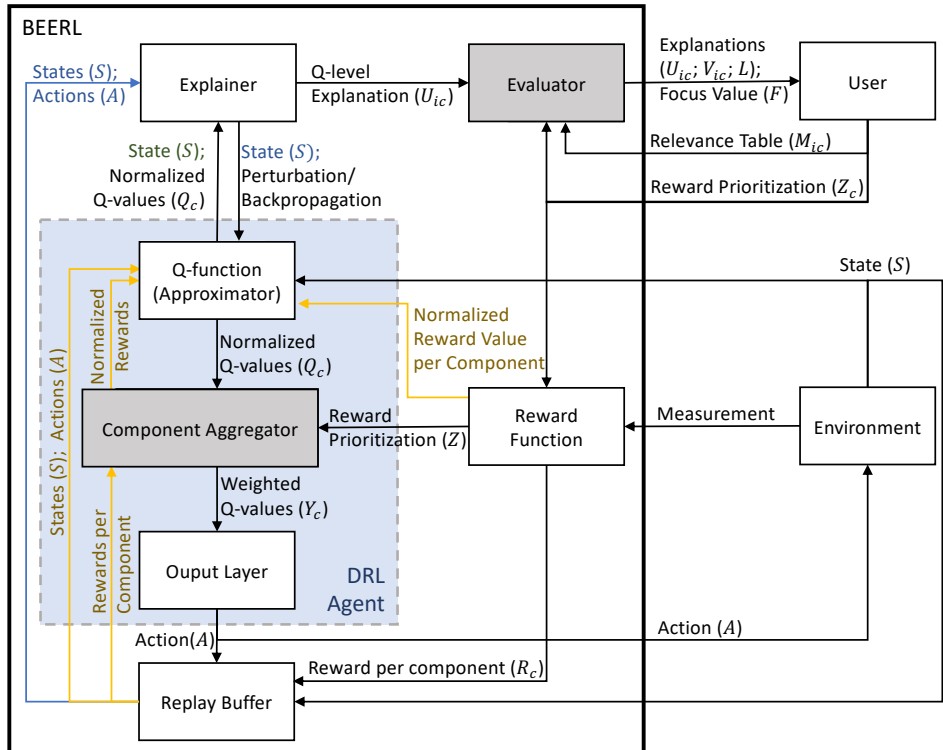

**Figure 1.** Main elements of BEERL and interactions among them. The gray boxes present the newly proposed elements in this work. The blue line is executed for generating a global explanation while the green text (state information) is passed when generating local explanation. The yellow lines are only executed during training process i.e., updating Q-function parameters.

### 3.1. RL Environment

An RL agent is deployed in a certain environment where it observes the state and performs its actions. In the proposed method, the environment should be constructed using several components to decompose the total reward calculation into different reward functions, as is performed in [7,8]. The state and action formulations do not need to be modified and are similar to the RL without reward decomposition.

### 3.2. Reward Functions

In a non-decomposed RL, the reward is provided as a single numeric value, while our method follows [7,8] to decompose the reward value into several components($r_c$). In our method, each reward component must be defined in a normalized manner, that is, within the same range (e.g., within the range $[-1, 1]$). The normalized rewards are fed to the Q-function during the training process to update Q-function parameters. Additionally, the output range of the Q-function is set at a normalized value to avoid the vanishing or exploding gradient of DNN parameters. It can happen when the output range is too large

or too small, as addressed in [37] and also exemplified in Lunar Lander in Appendix D.2. However, the weighted reward values are stored in a replay buffer in order to be summarized when a global explanation is generated. The summary of the reward component is calculated by taking the absolute mean of all the recorded reward components:

$$R_c = \frac{\sum_{t=1}^{T} |r_{ct}|}{N_T} \tag{1}$$

Throughout this paper, $i \in I$ indicates input feature and $c \in C$ indicates the reward component. Additionally, $N$ indicates the count of the referred annotation, e.g., $N_T$ annotates the number of time steps, $N_I$ annotates the number of input features, $N_C$ annotates the number of reward components, etc.

### 3.3. User Configuration

Reward prioritization is required to encode the magnitude of each component in calculating the total reward because the reward functions generate values in a normalized manner. Furthermore, our method also requires a relevance table, which encodes the desired behavior of the RL agent. In return, after completing the entire process, the user will receive three different types of explanation, which are (1) attributive explanations that consists of Q-function explanation ($U_{ic}$), RL explanation ($V_{ic}$), and normalized explanation ($L_{ic}$); (2) contrastive explanation ($\Delta V_{ic}$); and (3) focus values ($F_c$), which are explained in Sections 3.5 and 3.6.

#### 3.3.1. Reward Prioritization

We decouple reward prioritization ($Z_c(r)$) from reward function ($R_c(o)$) to generate two levels of explanation (Q-function and RL level). Thus, the normalized value is used as the target value for the NN that approximates the Q-value that the respective reward prioritization is then applied to configure the contribution of each reward component. $Z_c(r)$ may implement a complex formula (e.g., quadratic or exponential), while a linear function is preferred because it can be interpreted as a ratio among different reward components (e.g., $Z_{c1}(r) : Z_{c2}(r) : Z_{c3}(r) = k_{c1} : k_{c2} : k_{c3}$). Reward priorities are maintained during training and can be adjusted after completion for transfer learning purposes, as discussed in Section 4.6.

#### 3.3.2. Relevance Table

In designing an RL agent, the user commonly has expectations of how the agent will behave in performing its task. Normally, this expectation is encoded in a reward function where a fraction of the state variables contribute to a reward component, and another fraction of the state variables contributes to another reward component. Our method requires the user to define the relevance of each input feature (state variable) to each reward component in the form of a relevance matrix $M_{ic}$ where $i$ represents the input feature and $c$ represents the reward component. This matrix quantifies the desired behavior and the value of each element should be within the range of $[0, 1]$ where 1 quantifies the complete relevance of an input feature to a reward component.

### 3.4. RL Agent

The RL is formulated as Markov decision processes (MDPs) with a tuple of $(S, A, H, R, \gamma)$ where $S$ represents the state space, $A$ represents the action space, $H$ represents the transition function ($H(s, a)$), $R$ represents the reward function, and $\gamma$ represents the discount factor. The goal of the RL agent (as shown within the blue box in Figure 1) is to maximize the expected return ($J_t = \sum_{t=1}^{\infty} r_t \gamma_t$) when performing its task. A Q-function $Q^\pi(s, a)$ generates Q-values that measure the quality of the policy given the current state ($s$), and the action ($a$) executed on the policy $\pi$ is formulated as $Q^\pi(s, a) = \mathbb{E}_\pi[J_t | s_t = s, a_t = a]$. The component aggregator applies the reward priority, and the output layer combines all weighted Q-values to select the action as described in the following subsections.

### 3.4.1. Q-Function

When the reward is decomposed, the Q-function is defined per component $Q_c(s, a)$. In DRL, a reward component can be implemented as an independent neural network structure that shares only input and output layers. This architecture allows every component to be modular (i.e., a component can be transferred to or taken from another trained model) at a cost of a higher number of parameters. Alternatively, a branched network can also be implemented, where the shared initial layers can be randomly initialized network or taken from a trained model as the feature extractor. In addition to allowing transfer learning, the branching architecture also reduces the number of parameters for the NN model.

In the proposed solution, Q-functions are trainable and can be considered as hidden layers of the NN when DRL is used. The input layer takes the state of the observed environment when performing its task or the perturbation data from the explainer when generating the explanation. The output of this element is the approximated Q-values for each action per reward component. During training, the last hidden layer generates the prediction of Q-values to be calculated with a normalized reward to update the NN parameters. In this manner, the NN training operates at a normalized value and has the benefit of avoiding vanishing/exploding gradients.

### 3.4.2. Component Aggregator

This element connects the output/last layer with the second-last layer. It is worth noting that every node on every component's last-layer is connected only to one node on the output layer (instead of fully connected to all output nodes). The main function of this component is translating normalized Q-values($q_c$) to weighted Q-values($y_c$) and vice versa (for training). To calculate the weighted Q-values, it receives and stores the reward prioritization ($y_c = Z_c(q_c)$). This prioritization is another major difference of this component from the summation of the reward component in [8], where they do not explicitly weight/prioritize the components.

### 3.4.3. Output Layer

This element aggregates the weighted Q-values by taking a sum of all components.

$$Y = \sum_{c=1}^{C} Y_c \tag{2}$$

When the action space is continuous, these values are then performed as the RL action. Otherwise, if the action space is discrete, it selects the action that maximizes the reward in the exploitation phase. The selected action from this part will be performed in the environment to obtain the effect of it and also stored in the replay buffer for training purposes.

### 3.5. Generating Explanation

Our work generates two types of explanations, which are attributive and contrastive explanations. The former calculates the contribution towards the predicted output while the latter calculates the difference of two compared actions. Since the contrastive explanation compares two different actions, it is only produced in a local scope, whereas the attributive explanation is produced in both local and global scopes.

### 3.5.1. Attributive Explanation

Our presented approach is independent of the XAI method used to generate feature importance, i.e., any method (e.g., SHAP [4], LIME [5], etc.) can be applied. When a deep network is employed for the RL agent, the neural network explainer can also be applied to fit the required characteristics, e.g., the complexity of the operation, consistency, the accuracy of the explanation, etc. Without reward decomposition, the XAI methods generate a feature importance for the entire Q-function (i.e., the entire DRL network), as depicted in

Figure 4a. In the proposed solution, the explainer analyzes the Q-function by calculating the contribution of every input feature to every reward component:

$$U_{ic} = E(Q_c(s, a), s, a) \tag{3}$$

where $E(h, x, y)$ is the explainer that takes $h$ as the model to be explained (Q-function of reward component $c$, $Q_c(s, a)$) together with the input data $x$ and the predicted output $y$ (state $s$ and action $a$ in DRL, respectively). The generated attribution values, $U_{ic}$, explain the contribution in a normalized domain, which are independent of reward prioritization, and we refer to them as the Q-function explanations. Furthermore, we apply the reward prioritization to these explanations to generate the RL-level explanations ($V_{ic}$).

$$V_{ic} = Z_c(U_{ic}) \tag{4}$$

In this manner, the effect of reward prioritization is shown and each reward component can be individually analyzed. Any scope of explanations can be produced using this method, where the global explanation is an aggregation of local explanations using the absolute mean formula, as shown in Equation (5) ($P_{ic}$ refers to either level of explanation, i.e., $U_{ic}$ or $V_{ic}$).

$$P_{ic} = \frac{\sum_{t=1}^{T} |P_{ict}|}{N_T} \tag{5}$$

The above generated explanations show the importance of input feature towards different reward components, but currently, there is no mechanism to compare the importance of input features with reward components. With these detailed explanations ($U_{ic}$ and $V_{ic}$), the normalized input feature importance ($L_i$) is calculated by summing its importance to all reward components and dividing it by the number of reward components ($N_C$), as shown in Equation (6). Similarly, Equation (7) shows the calculation of normalized reward component importance ($L_c$) where all input feature importance are summed and then divided by the number of input features ($N_I$). In this manner, both input feature ($L_i$) and the reward component's ($L_c$) importances are measured on the same basis and can be properly compared.

$$L_i = \frac{\sum_{c=1}^{C} P_{ic}}{N_C} \tag{6}$$

$$L_c = \frac{\sum_{i=1}^{I} P_{ic}}{N_I} \tag{7}$$

### 3.5.2. Contrastive Explanations

The proposed method generates $N_C \times N_I$ of reasons because each explanation consists of a tuple of input feature and reward component. When comparing two actions, the contrastive explanation can be compressed by following the MSX concept [8]. We propose two possible compressions of the explanations. First, compression is performed by aggregating the reward component ($MSX^C$) resulting in explanations similar to the original MSX where only a subset of reward components' Q-values are highlighted, and second, we compress the MSX by aggregating the input feature ($MSX^I$). Details of the modified MSX calculations are presented in Appendix A.

Automatic compression is also proposed to generate the simplest contrastive explanations by implementing the Algorithm 1. We denote a set of compressed explanations by implementing the MSX concept as $MSX^{BEERL}$. The main task of this algorithm is to select one of $MSX^{BEERL}$, $MSX^I$, or $MSX^C$ to be highlighted. We first collect these MSX sets in the list of explanations(P). Second, the length of reasons (N) and the advantage values (A) of each MSX set are also stored. The advantage value is the difference between the total MSX contribution and the sum of all negative contributions, as formulated in Equation (A10). The selection is started by comparing the number of reasons in each MSX set and choosing the least if a single solution is obtained. When all MSX sets have the same number of

reasons, it selects the set with the highest advantage. When two sets of MSX have the same number of reasons, it removes a set of MSX with the most reasons and chooses the set with the highest advantage value between the two sets.

---

**Algorithm 1** Automatic MSX Compression

---

**Require:** $MSX^{BEERL}, MSX^I, MSX^C$
  $P \leftarrow [MSX^{BEERL}, MSX^I, MSX^C]$
  $N \leftarrow [N_{MSX^{BEERL}}, N_{MSX^I}, N_{MSX^C}]$
  $A \leftarrow [Adv(MSX^{BEERL}), Adv(MSX^I), Adv(MSX^C)]$   ▷ $Adv$ is advantage function (A10)
  $N_{min} \leftarrow min(N)$                     ▷ Check the fewest reason from the MSXs
  **if** $|N_{min} \in N| == 1$ **then**           ▷ only 1 MSX has the simplest explanation
    $d = \arg\min N$         ▷ Select the fewest reason for the final explanations
  **else if** $|N_{min} \in N| == 3$ **then**
    $d = \arg\max A$        ▷ Select the explanations with the highest advantage
  **else**
    $d = \arg\max N$     ▷ Select the explanations with most reasons to be removed
    pop($P_d$); pop($P_d$); pop($P_d$);       ▷ Remove the explanation from the lists
    $d = \arg\max A$        ▷ Select the explanations with the highest advantage
  **end if**
  **return** $M_d$

---

### 3.6. Evaluator

Humans evaluate the presented explanation mostly qualitatively. When an explanation consists of many elements, humans may be overwhelmed by the amount of information. On the other hand, no quantification of how the explanation satisfies the desired outcome is available in the existing work. We propose a focus value that evaluates the generated explanation by quantifying how much the explanation satisfies the desired behavior. The focus value ($F_c$) is the result of element-wise multiplication between the relevance table ($M_{ic}$) and the explanation of the Q-function ($U_{ic}$) then it is averaged per component.

$$F_c = \frac{\sum_{i=1}^I M_{ic} \odot U_{ic}}{\sum_{i=1}^I U_{ic}} \tag{8}$$

In addition, the unweighted and weighted mean values are also calculated. The unweighted mean value ($F_{unweighted}$) is merely the mean of all components:

$$F_{unweighted} = \frac{\sum_{c=1}^C F_c}{N_c} \tag{9}$$

while the weighted mean ($F_{weighted}$) considers the reward prioritization:

$$F_{weighted} = \frac{\sum_{c=1}^C Z_c(F_c)}{\sum_{c=1}^C Z_c(1)} \tag{10}$$

where $Z_c$ is the reward prioritization.

### 3.7. Data Flow

In generating local explanations, the explainer takes a state that is then modified to calculate the feature importance. On the other hand, all recorded states are taken by the explainer and are then fed to the Q-function to generate global explanations. The first generated explanation is the feature importance at Q-function level ($U_{ic}$ or Q-FI in Figure 2). The feature importance at the RL level ($V_{ic}$ or RL-FI in Figure 2) is generated by applying reward prioritization to the Q-level explanation. The focus values are then generated after applying focus value formulas (Equations (8)–(10)) to the explanations.

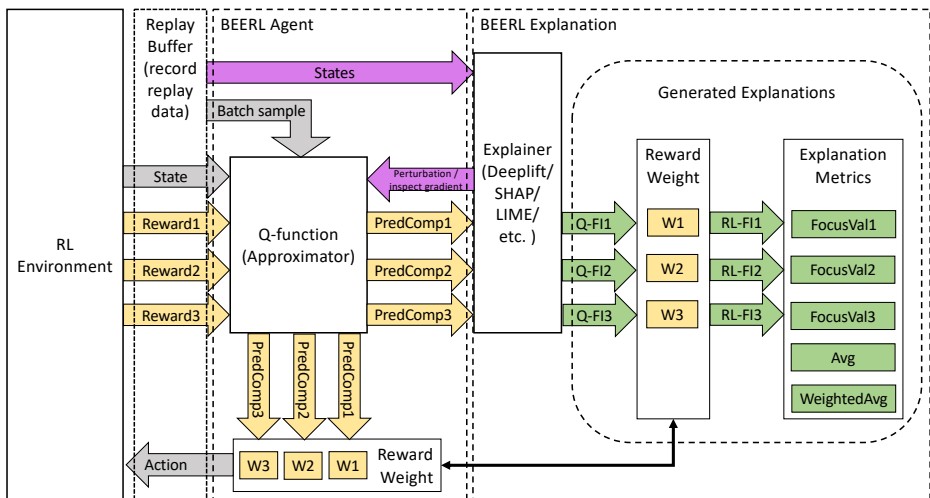

**Figure 2.** Illustration of BEERL dataflow in generating explanations in two different levels. The BEERL explanations are shown by green color while reward decomposition elements as introduced in [7] are shown by yellow; feature important components are shown by purple; common RL components are shown by grey.

## 4. Experimental Evaluations and Results

We experiment with a realistic use case, where DRL agents are used in a remote electrical antenna tilt environment, as shown in Figure 3. It is a multi-agent environment where each antenna is controlled individually. In this experiment, we trained a shared policy using DQN with reward decomposition (drDQN) where each agent obtains its observation, but uses the same policy as the others.

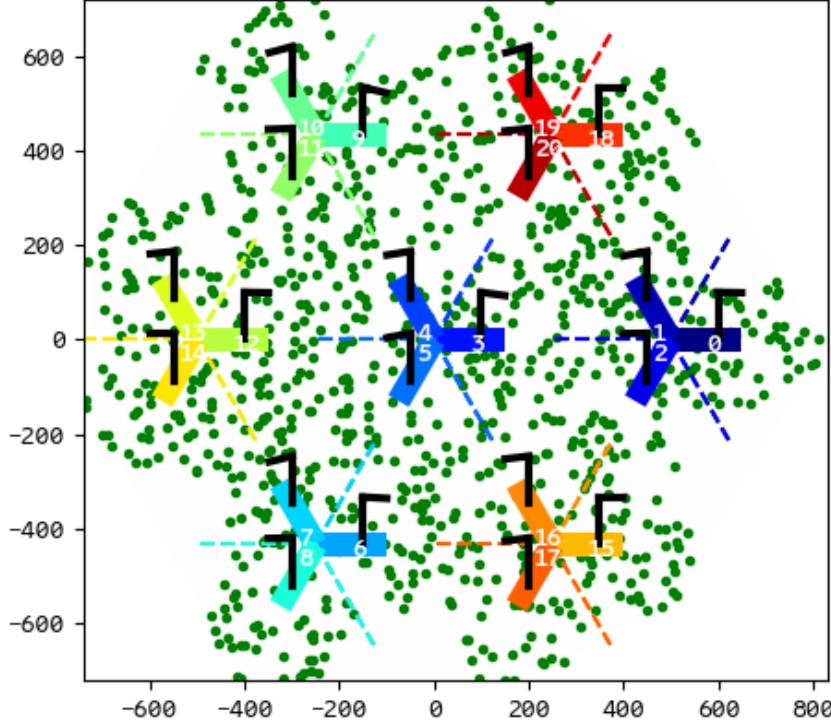

**Figure 3.** The simulation map with seven base stations arranged in a hexagonal shape where each of them has three antennas. The antennas are illustrated as inverted L-shape lines in black where their directions are illustrated in various colors. The green dots illustrated the position of the UEs that are spread across the field.

### 4.1. Remote Electrical Antenna Tilt Environment

Remote Electrical Antenna Tilt is a joint-optimization problem in which the agent controls the tilt of the antenna to optimize several key performance indicators (KPI). The antenna performance is evaluated mainly in three different aspects, namely the quality, capacity, and coverage of the traffic delivered by each antenna. The quality KPI is measured by the Signal-to-Interference and Noise Ratio (SINR), where the user's equipment (UE) signal strength is compared to noise and interference, and capacity KPI measures the throughput (Tput) delivered from the antenna to the UE, while the coverage is indicated by the Reference Signals Received Power (RSRP) showing the received power level of a reference signal. The challenge of this setup lies in the opposing KPIs that are used. When an antenna is fully tilted down, the signal beam covers a narrow spot, delivering high capacities due to fewer UEs being served, and high-quality signal due to less interference from the opposing antenna. However, when the antenna is fully tilted up, it will cover a wider area, but it will deliver low capacities to the UEs and low quality due to interference from other antennas. Therefore, it is important to understand the factors of the performing agent when controlling the tilt of the antenna.

In this work, the inputs (state variables) of the RL agent consist of the tilt position, RSRP, SINR, and throughput of the antenna. DeepliftSHAP [4] is applied to analyze the contribution of these input features to the prediction generated by the RL agent. The reward for the agent is constructed with factors similar to the input feature, which are SINR ($R_{SINR}$), RSRP ($R_{RSRP}$), and throughput ($R_{Throughput}$) metrics of the network. The reward calculation is shown in Equation (11) where the RSRP has twice the weight of other KPIs. This reward component has higher priority because the other two components have opposing properties, as mentioned above. We implement our drDQN model with three branches (one for each reward component) without shared layer, and each of them has two hidden layers where each layer has 32 nodes and rectified linear unit (ReLU) activation function.

$$R_{total} = R_{SINR} + R_{Throughput} + 2 \times R_{RSRP} \tag{11}$$

Once the RL training converged, we ran an evaluation where five episodes with 100 time steps in total, resulting in 2100 data points to collect the observations as the baseline for the Deeplift-SHAP explainer. Each datum is then fed to the trained agent to calculate the Q-values, select the optimal action, and generate a local explanation. The global explanation is calculated by taking the absolute mean value of all local explanations.

### 4.2. Explanations Using Existing XAI Methods

By implementing existing methods, we obtain the results shown in Figure 4. The importance of every input feature generated from DeepLiftSHAP [4] is shown in Figure 4a without showing its contribution to different components. We can see that the RSRP feature contributes significantly to the prediction of the agent, but there is no information on how it is distributed to the reward components. Similarly, Figure 4b shows the explanation of how each component constructs the total reward, as in [8] without showing the contribution of every input feature. For example, when the RSRP reward component ($R_{RSRP}$) contributes the most to the final reward calculation, the contributions of the input features are not shown.

### 4.3. Attributive Explanations with BEERL

Our method leverages the transparency of the last layers by implementing the reward decomposition method and separating the reward weight so that the reward components can be compared in a fair manner. This approach benefits in two aspects, which are as follows: (1) each reward component is trained in the same range, which minimizes the explosion or vanishing gradient; (2) the effect of reward prioritization is exposed so it helps humans understand the identification of unwanted behavior.

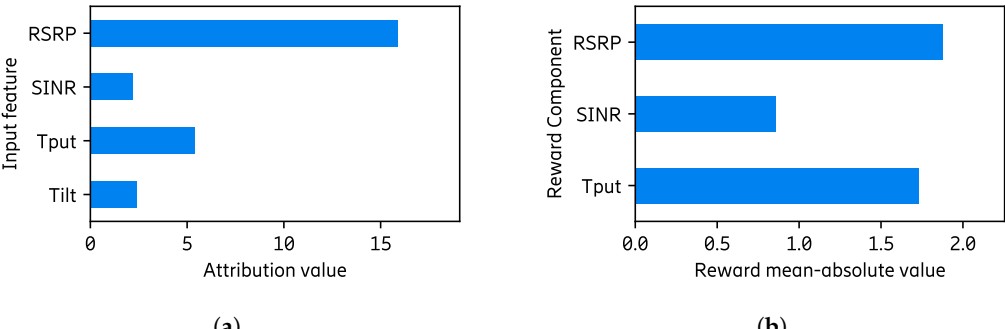

(**a**)           (**b**)

**Figure 4.** Explanations generated using the available XAI methods. (**a**) Feature importance generated using [6], which produces similar explanation format as [4,5]. (**b**) Reward components summary generated using [8].

Figure 5 shows the explanation on the Q-function level, i.e., how the Q-function (which is implemented in a neural network model) performs regardless of the priority of the reward component. From Figure 5a, we can see that the RSRP, SINR, and throughput input features have dominant importance to their respective reward components in comparison to other input features. Similarly, the respective input feature becomes the most important for each reward component (other input features do not outweigh their importance in every reward component), as shown in Figure 5b.

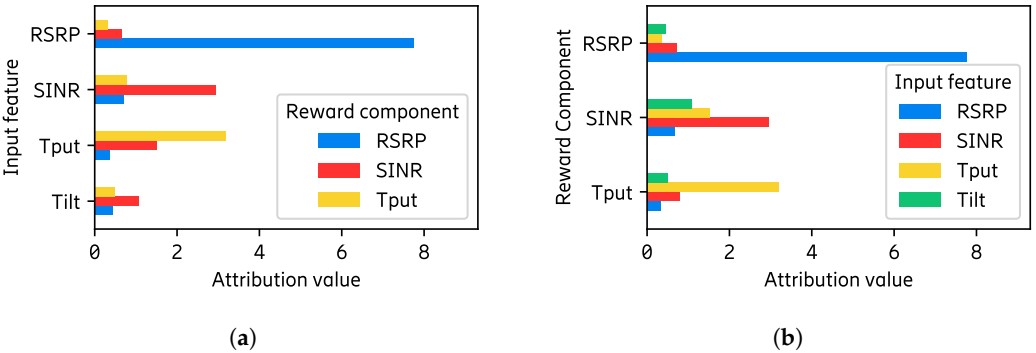

(**a**)           (**b**)

**Figure 5.** Explanations generated using BEERL at the Q-functions level without applying rewards prioritization. (**a**) Decomposed feature importance at the Q-function level. (**b**) Decomposed feature importance presented per reward component at the Q-function level.

When NN is employed for the RL agent, it is desirable that the Q-functions work in a normalized range to avoid the exploding or vanishing gradient phenomenon. Explanations at this level show the behavior of NN in approximating Q-values and the fairness of it (i.e., when all components are treated/trained in equal numeric values). Ideally, each feature has a significant contribution to at least one reward component. It is a waste of resources when an input feature is used, but it does not have any impact on the output. Similarly, each reward component should have at least an input feature that significantly affects it.

The explanation at the RL agent level shows the correlation between the input and output explanations, where the priority/weights of the reward components are considered. In this manner, we can clearly see which aspect affects the final action of the agent (e.g., whether the neural network as the Q-function, the reward prioritization, or the combination of them). When reward weights are applied and the RSRP reward component is prioritized more than others ($2.0 : 1.0 : 1.0$ for $k_{RSRP} : k_{SINR} : k_{Throughput}$), we can see that each input feature still dominates its importance for the respective reward component and vice versa, as shown in Figure 6. Unlike Q-function explanations, an expert may accept RL explanations where a reward component does not contribute significantly to the total reward. The contribution of the reward component can be adjusted using reward

prioritization, and the proportion of reward prioritization can be justified by the domain knowledge expert.

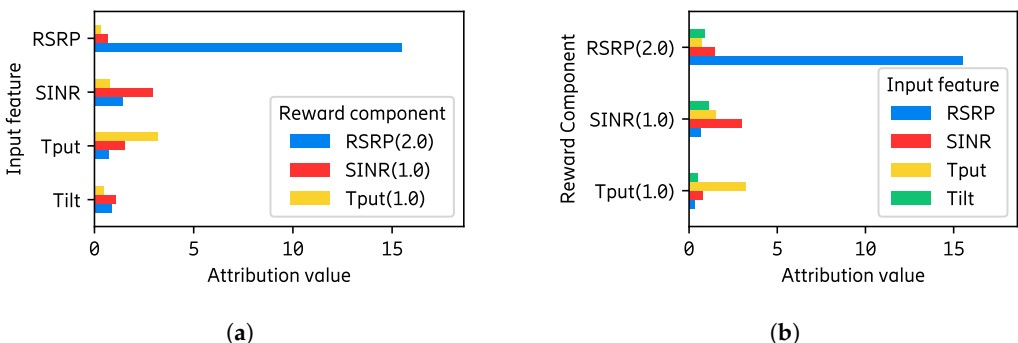

(**a**)                                                                                          (**b**)

**Figure 6.** Explanations generated using BEERL at the RL level where the respective reward prioritization (k; shown in parentheses) is applied to the every component. (**a**) Decomposed feature importance at the RL level. (**b**) Decomposed feature importance presented per reward component at the RL level.

The detailed feature importance is shown in Figures 5 and 6 without comparing the importance between the input features and the reward components. Figure 7 shows the importance of them in the same plot so that they can be compared which is also known as a normalized explanation ($L_{ic}$). We can see from Figure 7a that the RSRP input feature and the reward component become the top-two most important element in this agent. When reward weights are applied, their importances are pushed further, as shown in Figure 7b.

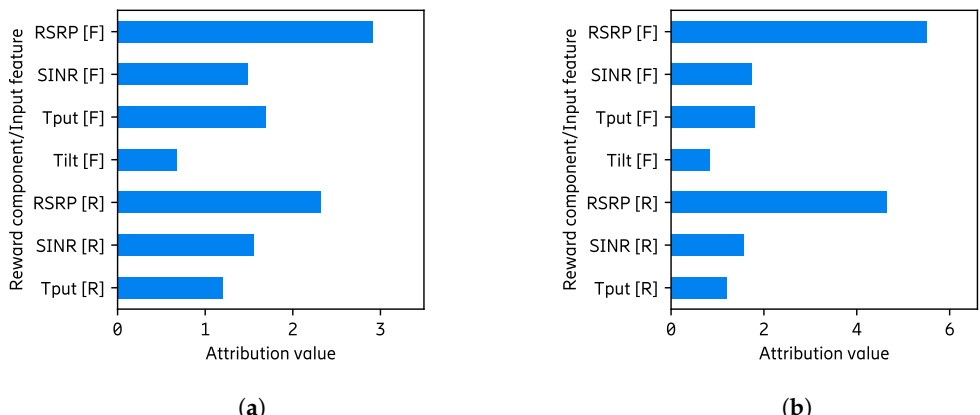

(**a**)                                                                                          (**b**)

**Figure 7.** The normalized explanations where the importance of the input features and the reward components are compared in the same plot. (**a**) Normalized explanation at the Q-function level. (**b**) Normalized explanation when the reward weight is applied (at RL-level).

In a local scope (i.e., explanation of a single action), the detailed information produced by our method exposes the contribution that has not been seen in existing methods. For example, information about negative contribution of RSRP and SINR input features to the throughput reward component cannot be inferred from feature importance or reward decomposition methods alone. This information is important, especially when several input features have opposing contributions. With an existing method, a feature may not be described as important because the attribution value is small or zero. However, it may actually contribute to different components in an opposing way (e.g., high positive contribution to reward component 1 and high negative contribution to reward component 2), as shown by SINR input feature importances on Figure 8 (top plots, second row). By using our method, hidden information, as mentioned, can be revealed and the user may have a better understanding of the performing agent.

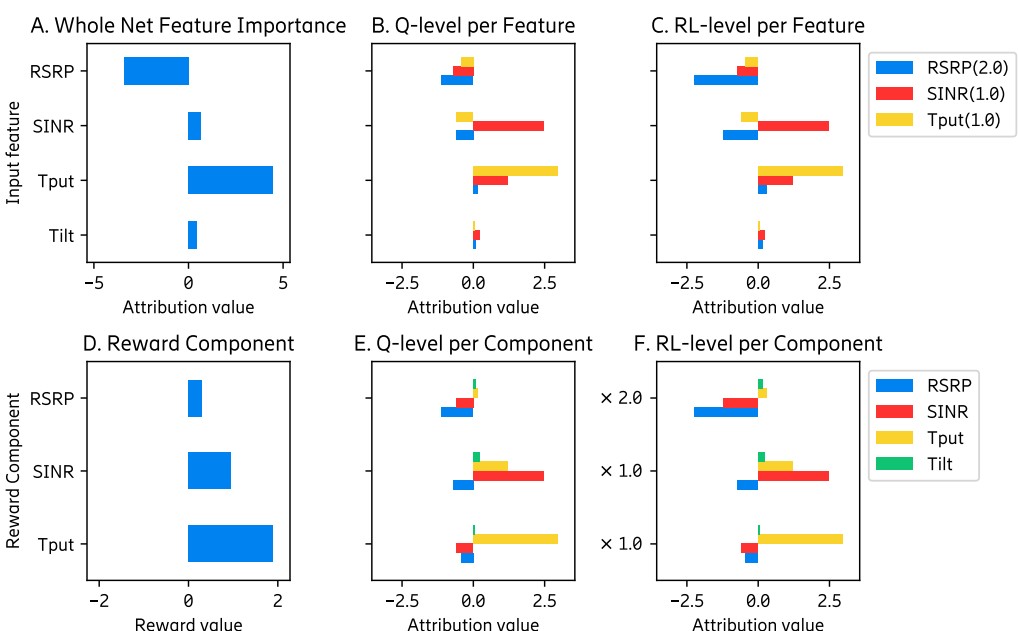

**Figure 8.** The local explanations of one antenna when the agent chose no-tilt action.

### 4.4. Analysis of Shapley-Based Explainer

If we sum the attribution values of input features towards the RSRP reward component on Figure 8 (bottom-right), we will obtain a negative value of the RSRP reward component. However, the predicted Q-value of this component is positive, as shown as a black bar on Figure 9b. When the explainer is built on the Shapley-concept (e.g., SHAP and its variants), the sum of attribution (i.e., feature importance) values equals to the output (or prediction) of the performing model. This value includes the baseline value, which is the average model output of the given dataset, where we present it as the purple bar in Figure 9b. In this manner, we have a complete explanation in which the predicted Q-value (black bar) is equal to the sum of all input features contributions including the base value of the reward component.

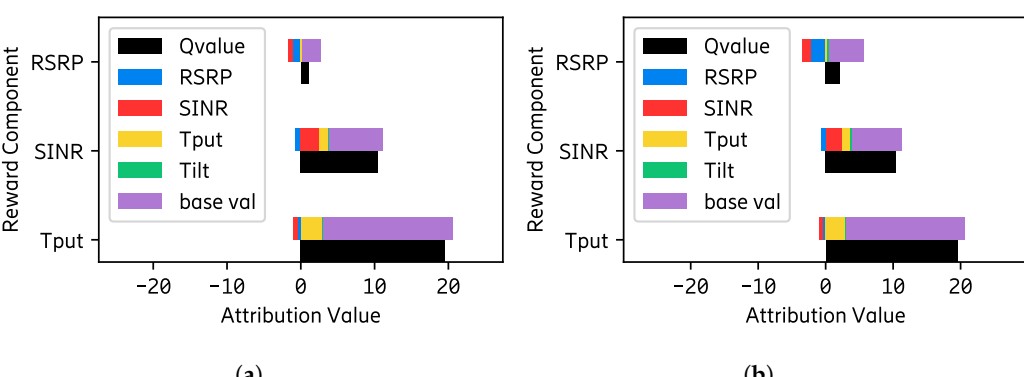

|  (a)  |  (b)  |

**Figure 9.** Complete explanation where the sum of feature attribution and base values equals the predicted Q-values. (**a**) Attribution and Q-values at the Q-level. (**b**) Attribution and Q-values on RL level.

In an implementation without decomposition, the baseline values are different for possible actions (or classes in a supervised learning problem). Thus, we can consider this baseline value as the bias towards a different action. In a model with reward decompositions, the difference in the base value appears not only for different actions but also for reward components, as shown by Table 1. Although these values are constant for any instance/state/observation, they play an important role in selecting the action. The differ-

ences in the base value among different actions are not as large as the difference between different reward components, as shown by the range on Table 1 (the range is the subtraction of the maximum by the minimum value of each reward component). However, the mean values show the average base value of each reward component, where the SINR has the lowest average base value and the throughput component has the highest value.

**Table 1.** Base value for each action and reward component at the RL level.

| Action | RSRP | SINR | Throughput |
|---|---|---|---|
| Tilt-down | 12.522070 | 8.453173 | 17.904276 |
| No-tilt | 12.652522 | 9.103990 | 18.448515 |
| Tilt-up | 12.470411 | 8.746243 | 18.138926 |
| Mean | 12.548335 | 8.767802 | 18.163904 |
| Range | 0.182111 | 0.650817 | 0.544239 |

When analyzing a local explanation, it is important that the explanation be accompanied by the base value as a reference to show all contributions leading to the predicted Q-value. In this manner, the explanation shows how each input feature impacts the model's output from the average situation. Although understanding why a specific base value is obtained remains limited to the fact that it is the average output value of the model from the given dataset, exposing it informs the complete analysis of the model. The dataset becomes the scope or corridor for generating the explanation. In connection with the normalized explanations, in Figure 7b, the information shows the importance of different model elements in terms of the variation of input values regardless of the bias of the reward component. In connection with the RL level explanation Figure 6b, the bias of the reward component may be overcome by the contribution made by the input feature(s). Specifically, for the experiment above, the mean bias (base value) for the throughput reward component (18.164) is higher than the RSRP (12.548). However, the mean absolute contribution of the RSRP input feature to the RSRP reward component is 15.501 where the sum of it and its bias (28.049) can overcome the throughput bias. This explanation shows that the model has bias towards different components, and the variation of the input values may overcome it.

*4.5. Contrastive Explanations with BEERL*

In the exploitation phase, an RL agent chooses an action that maximizes the cumulative reward, and understanding why it is preferred over the other is important. When contrasting the possible actions (e.g., why no tilt is preferred than tilting up), the generated explanations from our method present detailed information as shown in Figure 10 where they consist of input feature and reward component tuples. Figure 10a shows the explanation of why no tilt is preferred than tilting down where the tilt input feature contributes negatively to all reward components. On the other hand, the sum of positive contributions is greater than the disadvantage of not tilting up. When the $MSX^+$ concept is applied, simpler explanations are produced, such as the following: the contribution of '*RSRP[I] to SINR[R]*' and '*base value[I] to SINR[R]*' ([I] and [R] indicate input feature and reward component, respectively) are enough to overcome the disadvantage of not selecting action tilting down (shown as textured bar in Figure 10). Since both reasons contribute to the SINR reward component, the Algorithm 1 further compresses it by aggregating the contribution of each reward component($MSX^C$). Thus, the compressed explanation can be interpreted as '*The contribution of SINR reward component is enough to overcome the negative impact of not choosing tilting down action*'.

Figure 10b shows another constrastive explanation of why no tilt is preferred than tilting up. The $MSX^+$ of these explanations informs that the contribution of '*Tilt[I] to AvgSINR[R]*', '*Tilt[I] to AvgRSRP[R]*', '*base val[I] to AvgSINR[R]*', and '*base val[I] to AvgThroughput[R]*' are enought to overcome the negative contribution of not tilting up. Furthermore, they are compressed by aggregating the contribution of the input feature resulting

in an explanation that can be interpreted as '*The contribution of Tilt input feature and base values are enough to overcome the negative impact of not choosing tilting up action*'. It should be noted that the base values in Figure 10 are the difference among the actions that the reward weight adjustment does not address. Adjusting the reward weight cannot change the proportion of it within each reward component (i.e., making it less dominant than the input features).

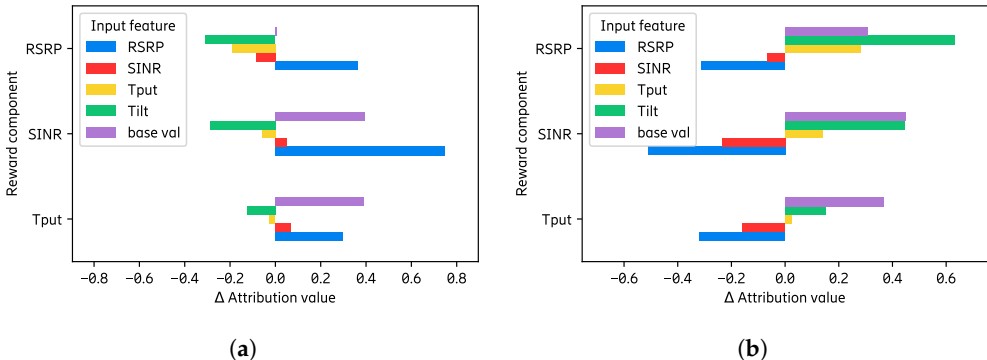

(**a**)                                      (**b**)

**Figure 10.** Contrastive explanations generated using BEERL where the MSX is highlighted. (**a**) Contrastive explanations about why no tilt action is chosen rather than tilting down. (**b**) Contrastive explanations about why no tilt action is chosen rather than tilting up.

### 4.6. Usages with BEERL

By presenting the explanations as above, we can gain more insight about the trained agent than the available methods by noting some findings, such as the following:

- The RSRP input feature is the most contributing feature and significantly affects the RSRP reward component. This explanation shows in detail the contribution of each input feature to every output component and how they are correlated, as shown in Figures 5a and 6a.
- SINR and throughput reward components are correctly focused on SINR and throughput input features, respectively. It shows that the Q-function performs well in focusing on the respective feature.
- The SINR and throughput reward components have less contribution than RSRP reward component even on the Q-function level. Applying the reward weights pushes the contribution further, where RSRP has a significantly larger contribution than other components.
- The base values show the bias of different actions and reward components. Even though the RSRP reward component is weighted twice that of others, the throughput reward component remains as the highest. This means that a higher contribution from the input feature(s) is required if other component(s) need to be overruled in selecting the action.
- If there is a situation where the priority of the reward components needs to be adjusted, one can reuse the trained model and then only adjust the reward weight and continue the training without retraining it from scratch with a random policy. With the given explanations, we are confident that the model is performing well because each reward component focuses on the correct features. This information presents the correctness of the proposed method, which not only increases human trust in the AI used but also allows for the transfer of knowledge with stronger reason.

From the above points, the explanations improve human knowledge to understand the behavior of the model in different elements of the RL model. The effect of the reward weight, the bias of actions and the reward component, and the contribution of the input feature and the reward components are produced to inform the detailed properties of the model's behavior. An indication of a better understanding is that the agent can be adjusted or reconfigured to fit the necessary goal. In the following, we show the adjustment

of the reward weights to reduce the bias (difference of the base value) among different reward components.

As the mean base values at the Q-function level are 6.274168, 8.767802, and 18.163904 for the RSRP, SINR, and throughput reward component, respectively, we adjusted the reward weight to achieve similar values. We choose an arbitrary value (in this case, it is 10) that we divide by the mean values of the base values of the reward components. Therefore, to keep two decimal values, we set the new reward weight at 1.60, 1.14, and 0.55 for RSRP, SINR, and throughput, respectively. Subsequently, we conducted an evaluation process and obtained the explanations, including bias, as shown in Figure 11 and Table 2.

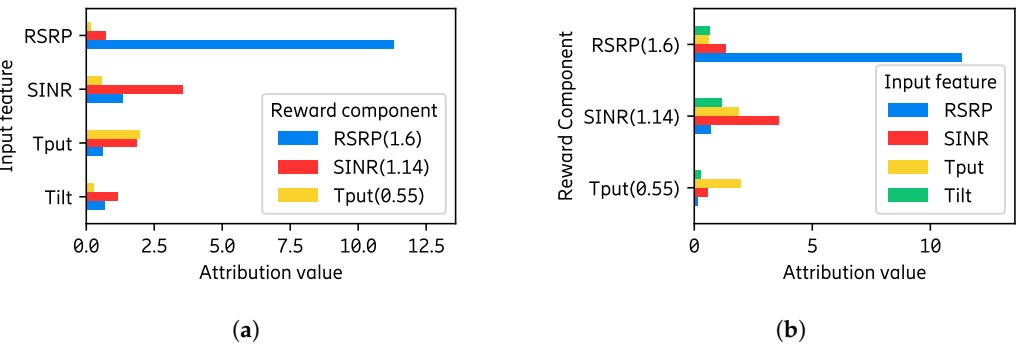

(**a**)                                          (**b**)

**Figure 11.** Explanations generated using BEERL at the RL level after the weights of the reward components (shown in parentheses) are adjusted. (**a**) Decomposed feature importance at the RL level. (**b**) Decomposed feature importance presented per reward component at the RL level.

**Table 2.** Base value for each action and reward component at the RL level after the reward weights are adjusted to reduce the bias of the reward components.

| Action | RSRP | SINR | Throughput |
|---|---|---|---|
| Tilt-down | 9.273874 | 9.247864 | 9.758694 |
| No-tilt | 9.375855 | 9.965568 | 10.043088 |
| Tilt-up | 9.231198 | 9.528765 | 9.867805 |
| Mean | 9.293643 | 9.580732 | 9.889862 |
| Range | 0.144657 | 0.717704 | 0.284394 |

As the reward adjustment is made relative to the trained agent, the explanations on the Q-function level are similar to Figure 5. From Figure 11, we can see that the throughput input feature has a relatively similar contribution to the SINR and throughput reward components. The importance of the characteristics in each reward component is also changed following the updated reward weights, as shown in Figure 11b. Table 2 shows the base values after the reward weights were adjusted, resulting in a reduction in the bias of the reward components. This weight adjustment exemplifies the possibility of reconfiguring the RL agent after training to achieve the desired properties.

Furthermore, we evaluated antenna telecommunication metrics (RSRP, SINR, and throughput) before and after adjusting the agent reward weights as shown in Figure 12. We can see that the agent with the adjusted reward weights has a better RSRP KPI than the original agent, while SINR and throughput KPIs are slightly compromised. Since the range of the base values of the adjusted agent is low, the action selection can depend more on the contribution of input features to the reward components and less on the base values of the reward components.

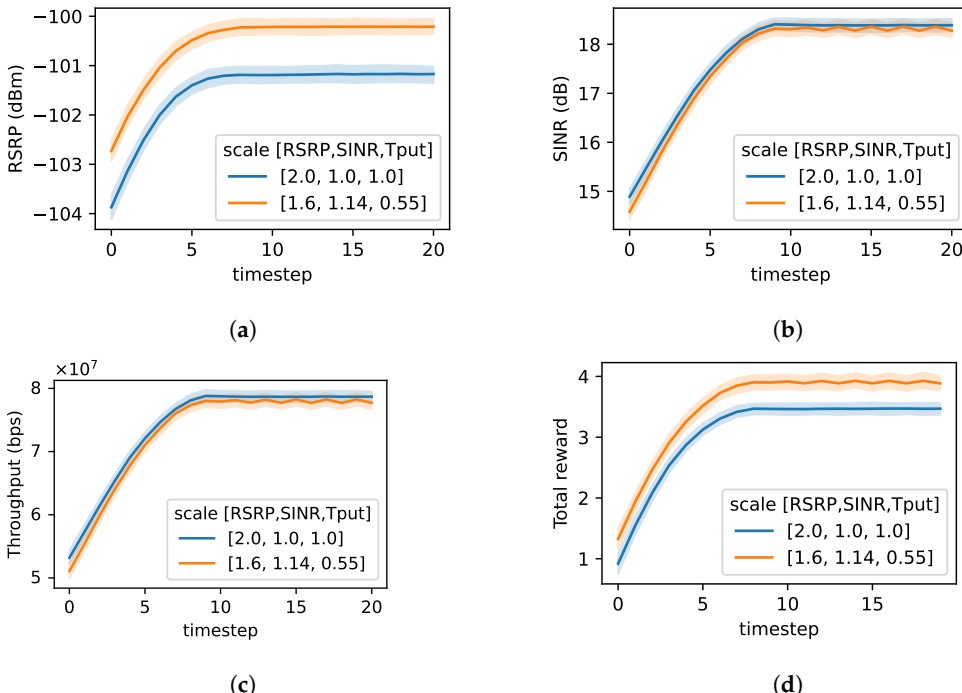

**Figure 12.** The average network metrics from fifty episodes showing the performance of the agent with original (blue) and adjusted reward weights (orange). (**a**) The RSRP signal on the evaluation phase. (**b**) The SINR signal on the evaluation phase. (**c**) The throughput signal on the evaluation phase. (**d**) The total reward on the evaluation phase.

*4.7. Focus Value: Quantifying the Desired Propreties*

To measure the fulfilment of the generated explanation with the desired properties as calculated by the focus value formula Equation (8), the user must set a relevance table ($M_{ic}$). In this experiment, the configured relevance table is presented in Table 3 with the rationale explained in the following sentences. The RSRP, SINR, and throughput reward component are highly correlated with the RSRP, SINR, and throughput input features, respectively. The tilt input feature also has a high correlation because it directly affects the antenna performance in all aspects. However, RSRP, SINR, and throughput are also correlated with each other in a moderate manner. Therefore, we set the correlation value at 0.5 since we want each reward component to focus on its most correlated input feature. The relevance table can be set by the user as she/he formulates the desired properties, and the rationale of the given example (Table 3) is explained above.

**Table 3.** Relevance table configured by the user quantifying the desired properties.

| Input | Reward Component | | |
|---|---|---|---|
| **Feature** | **RSRP** | **SINR** | **Throughput** |
| Tilt | 1.0 | 1.0 | 1.0 |
| RSRP | **1.0** | 0.5 | 0.5 |
| SINR | 0.5 | **1.0** | 0.5 |
| Throughput | 0.5 | 0.5 | **1.0** |

The generated explanations (Figures 5 and 11) are evaluated with the relevance table (Table 3) to generate the focus values shown in Table 4. The focus values for both experiments are relatively similar because adjusting the weight does not affect the proportion of the input features contribution to each reward component. The RSRP reward component has the highest focus value because the RSRP input feature has a significant contribution compared to other features. On the other hand, the SINR reward component has the

lowest focus value because it has significant contributions from the tilt and throughput input features.

**Table 4.** Focus values of the agent with original and adjusted reward weight, which quantifies how much the desired properties are fulfilled.

| Focus | Reward Component | | | Weighted | Unweighted |
|---|---|---|---|---|---|
| Value | RSRP | SINR | Throughput | Mean | Mean |
| Original weight | 0.942084 | 0.824311 | 0.884382 | 0.898215 | 0.883592 |
| Adjusted weight | 0.929772 | 0.823575 | 0.878465 | 0.884397 | 0.877271 |

The weighted mean has a higher value than the unweighted mean focus value, which means that reward prioritization mitigates unwanted behavior because components with high focus values have a higher priority. On the other hand, the unweighted focus value is lower, indicating that it has less desired behavior if we treat all components equally (i.e., do not apply prioritization).

## 5. Conclusions

This paper presents a novel "Both Ends Explanations for RL (BEERL)" method that generates fine-grained explanations, which connects and compares the input and output explanations of a black-box model. In this manner, the contribution of every feature to every reward component is revealed explicitly, allowing each element to be evaluated individually. The explanations are generated at two different levels as a result of exposing reward priority. The first is at the Q-level, where all components generate values in the same range to minimize the risk of vanishing or exploding gradients, as exemplified in Appendix D.2. The second is at the RL-level where the reward priority is applied to explain the rationale behind choosing an action. Our method also produces a detailed contrastive explanation that explains why one action is chosen instead of another. Additionally, the contrastive explanation can be compressed to highlight the important element(s) that make it more comprehensible to humans. To identify the most important factor of the agent, the input feature and the reward component are compared by aggregating their contributions. The implemented agent can be modified by adjusting the reward priority to fulfill the desired properties, such as reducing the bias of the reward components. A further adjustment of the RL agent can also be performed, such as retraining the Q-function partially per component, removing non-contributing input feature(s), removing misbehaving or suboptimal reward component(s), and tuning reward prioritization without restarting the training process, etc.

We implemented our method with respect to the remote electrical antenna tilt use case (as described in Section 4) and to two openAI gym environments (see Appendix D) to demonstrate the benefits of resultant explanations. A fine-grained explanation implies that the amount of information received by the user increases. Using the relevance table set by the user, the focus value summarizes the fulfillment of the desired properties. In the future, we intend to investigate a method to improve the focus value of the RL agent. This can be performed by incorporating the explanation during the training process. More experiments can be conducted on top of this work to mitigate bias between different actions.

## 6. Patents

A patent with the title "Explaining Operation of a Neural Network" has been filed as a US provisional application on 11 January 2022.

**Author Contributions:** Conceptualization, A.T., R.I. and E.F.; methodology, A.T.; writing—original draft preparation, A.T.; writing—review and editing, A.T., R.I. and E.F.; visualization, A.T.; supervision, R.I. and E.F. All authors have read and agreed to the published version of the manuscript.

**Funding:** This work was partially supported by the Wallenberg AI, Autonomous Systems and Software Program (WASP) funded by the Knut and Alice Wallenberg Foundation. (Grant number: EAB-22:009816).

**Institutional Review Board Statement:** Not applicable.

**Informed Consent Statement:** Not applicable

**Data Availability Statement:** Not applicable.

**Conflicts of Interest:** The authors declare no conflict of interest.

## Abbreviations

The following abbreviations are used in this manuscript:

| | |
|---|---|
| AI | Artificial intelligence; |
| BEERL | Both ends explanations for reinforcement learning; |
| DNN | Deep neural network; |
| DQN | Deep Q-network; |
| DRL | Deep reinforcement learning; |
| HRA | Hybrid reward architecture; |
| KPI | Key performance indicator; |
| MDP | Markov decision process; |
| MSX | Minimal sufficient explanation; |
| NN | Neural network; |
| ReLU | Rectified linear unit; |
| RL | Reinforcement learning; |
| RSRP | Reference Signals Received Power; |
| SINR | Signal to Interference and Noise Ratio; |
| Tput | Throughput; |
| UE | Users equipment; |
| XAI | Explainable artificial intelligence; |
| XRL | Explainable reinforcement learning. |

## Appendix A. MSX Formulation

As our method generates more detailed explanations, the calculation of MSX elements is adjusted to incorporate the input feature's attribution. The first calculation that is required is the RDX calculation ($\Delta(s, a_1, a_2)$), which is adjusted the following.

$$\Delta(s, a_1, a_2) = Z(E(Q(s, a_1), s, a_1)) - Z(E(Q(s, a_2), s, a_2)) \tag{A1}$$

The disadvantage value (sum of the negative elements of RDX, *d*) is then also adjusted to the following.

$$d = \Sigma_{ci} I[\Delta_{ci}(s, a_1, a_2) < 0] \cdot |[\Delta_{ci}(s, a_1, a_2) < 0]| \tag{A2}$$

Thus, due to the incorporation of feature attribution, in this work, $MSX^+$ becomes the following.

$$MSX^+ = \underset{M \in 2^{CI}}{\arg\min} |M| s.t. \sum_{c,i \in M} \Delta_{ci}(s, a_1, a_2) > d \tag{A3}$$

We further compress the $MSX^+$ by the reward component ($MSX^{+C}$), which results the same equation as the original $MSX^+$.

$$MSX^{+C} = \underset{M \in 2^{C}}{\arg\min} |M| s.t. \sum_{c \in M} \Delta_c(s, a_1, a_2) > d \tag{A4}$$

Alternatively , it is also possible to compress the $MSX^+$ by the input feature ($MSX^{+I}$), which formulated as follows.

$$MSX^{+I} = \underset{M \in 2^{I}}{\arg\min} |M| s.t. \sum_{i \in M} \Delta_i(s, a_1, a_2) > d \tag{A5}$$

To adjust $MSX^-$ with our implementation, the just-insufficient value ($v$) is modified to following.

$$v = \sum_{c,i \in MSX^+} \Delta_{ci}(s, a_1, a_2) - \min_{c,i \in MSX^+} \Delta_{ci}(s, a_1, a_2) \tag{A6}$$

Then, the $MSX^-$ is also adjusted:

$$MSX^- = \arg\min_{M \in 2^{CI}} |M| \text{ s.t. } \sum_{c,i \in M} -\Delta_{ci}(s, a_1, a_2) > v \tag{A7}$$

where it can also be compressed by the reward component ($MSX^{-C}$) or by the input feature ($MSX^{-I}$).

$$MSX^{-C} = \arg\min_{M \in 2^C} |M| \text{ s.t. } \sum_{c \in M} -\Delta_c(s, a_1, a_2) > v \tag{A8}$$

$$MSX^{-I} = \arg\min_{M \in 2^I} |M| \text{ s.t. } \sum_{i \in M} -\Delta_i(s, a_1, a_2) > v \tag{A9}$$

Additionally, we propose that the advantage value (net contribution) is then also needed.

$$a = \Sigma_{ci} I[\Delta_{ci}(s, a_1, a_2) > 0] \cdot |[\Delta_{ci}(s, a_1, a_2) > 0]| - d : c, i \in MSX^* \tag{A10}$$

## Appendix B. RL Parameters

The RL agent's parameters for the antenna tilt use case are shown in Table A1.

**Table A1.** The parameters of the RL agent for antenna tilt use case.

| Parameter | Value |
|---|---|
| Branch | 3 |
| Hidden layer | 2 |
| Shared layer | 0 |
| Node in hidden layer | (32, 32) |
| Activation function | ReLU |
| Max steps per episode | 20 |
| Training timesteps | 10,000 |
| Learning rate | $10^{-3}$ |
| Training batch | 64 |
| Initial epsilon | 1.0 |
| Final epsilon | 0.01 |
| Epsilon timesteps | 3000 |

## Appendix C. Antenna Tilt Configuration

The configuration of the antenna tilt experiment is presented in Table A2.

**Table A2.** The configuration of the antenna tilt environment.

| Parameter | Value |
|---|---|
| Number of base stations | 7 |
| Number of antennas | 21 |
| Number of user equipments | 1000 |
| Intersite-distance | 500 m |
| Frequency | 2.1 GHz |

## Appendix D. BEERL on Other Environments

In this section, we present the implementation of our method in the openAI environment that is publicly available for experimentation with RL problems [38]. The intention is to demonstrate that our method is applicable to other domains, and different insights can also be obtained from the following subsection.

### Appendix D.1. Cartpole from OpenAI Gym

In this environment, a simple control problem is simulated where a pole on a cart needs to be balanced so it can stand vertically by adjusting the cart in a horizontal direction. The state of this environment consists of the following: cart position($pos\_c$), cart velocity($vel\_c$), pole angle($\omega\_p$), and pole angular velocity($\theta\_p$). Since the cart is moving horizontally, there are two possible actions that push the cart to the left or right.

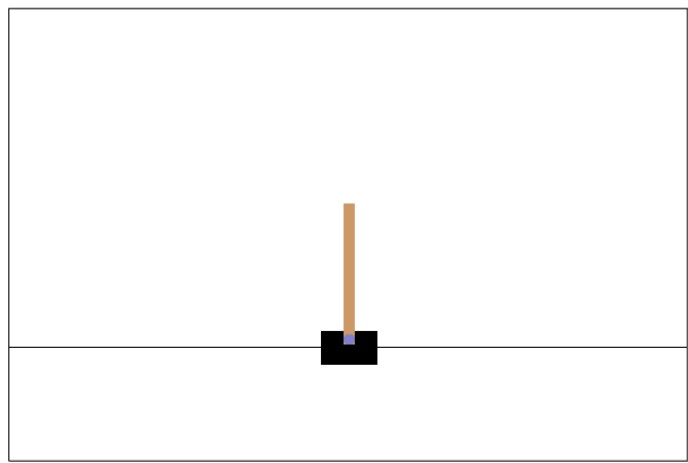

**Figure A1.** The illustration of the Cartpole environment from openAI gym.

The original reward was calculated by giving the $+1$ reward in each time step when the pole is upright. Otherwise, zero rewards will be given if the pole falls. However, to accommodate the decomposition of the reward, we modify the reward calculation as follows:

- Cart position reward: $R_{c\_pos}(s) = 1.0 - |\frac{pos\_c}{workspace_{cart}}|$;
- Pole angle reward: $R_{p\_\theta}(s) = 1.0 - |\frac{\theta\_p}{workspace_{pole}}|$;
- Total reward: $R_{total}(s) = k_{c\_pos} \times R_{c\_pos}(s) + k_{p\_\theta} \times R_{p\_\theta}(s)$.

We set the reward weight as 0.2 and 0.8 for the cart position reward ($k_{c\_pos}$) and pole angle reward ($k_{p\_\theta}$), respectively. The higher weight for the pole angle reward is given because it is the main task of this environment. In this manner, the agent is trained not only to keep the pole upright but also to position the cart at the center of the work area.

The global explanations of the agent in this environment are shown in Figure A2. At the Q-function level, all input features other than the position of the cart ($pos\_c$) have higher contributions to the position of the cart ($c\_pos$) than the pole angle's ($p\_\theta$) reward component. However, after applying the reward prioritization, the pole angle reward component dominates the contribution at the RL level. We can also see that the cart and pole angular velocity's ($vel\_c$ and $\omega\_p$) input features are the two most important features. Both have significant contributions to both the cart position ($c\_pos$) and pole angle ($p\_\theta$) reward components. When we compare the contribution of all input features and the reward components, as shown in Figure A3b, both the velocity input features ($vel\_c$ and $\omega\_p$) and the $p\_\theta$ reward component are the most contributing elements in the trained agent.

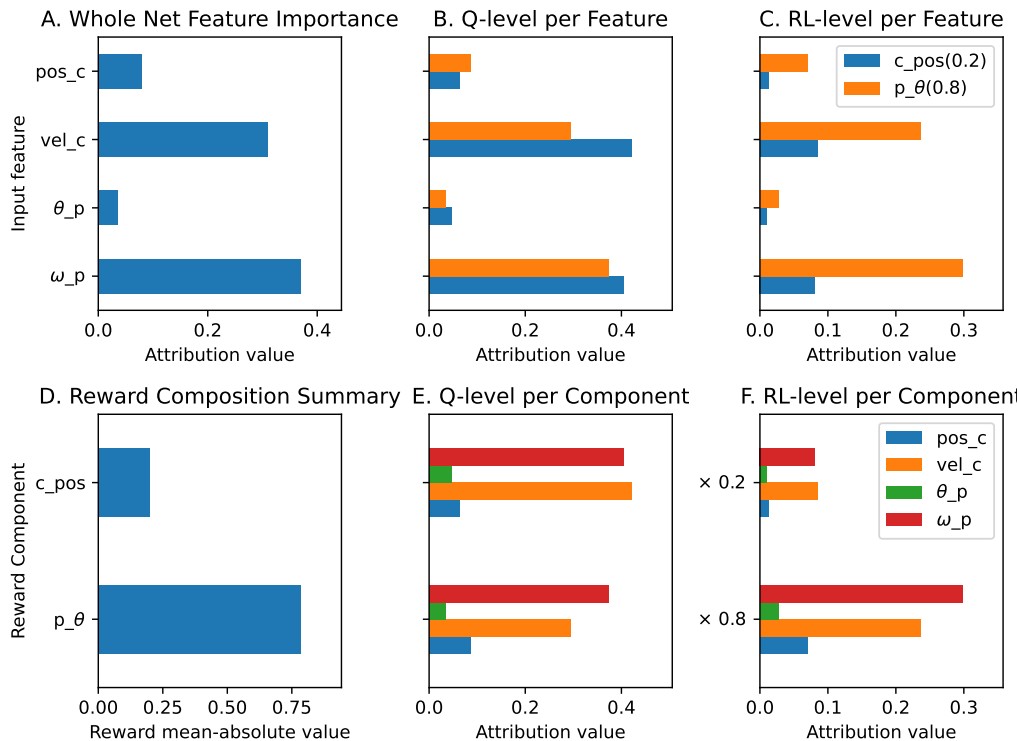

**Figure A2.** The global explanations of the cartpole agent.

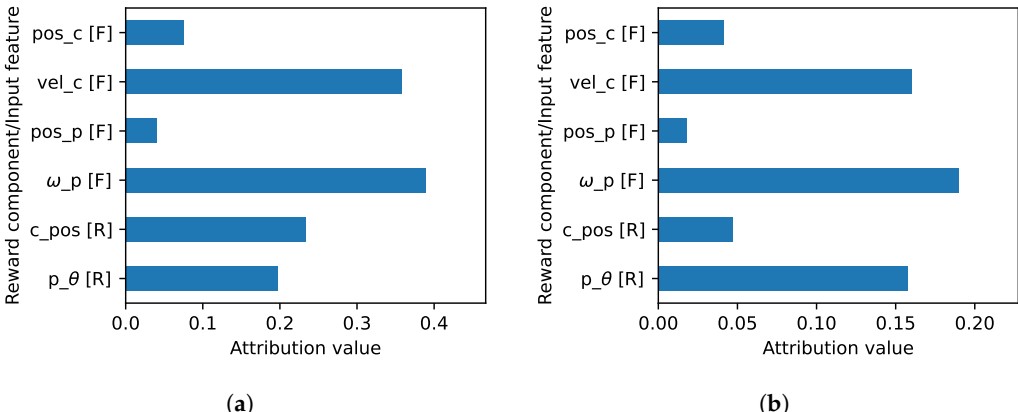

(**a**)          (**b**)

**Figure A3.** The normalized explanations of the cartpole agent where the importance of the input features and the reward components are compared in the same plot. (**a**) Normalized explanation at the Q-function level. (**b**) Normalized explanation when the reward weight is applied (at RL-level).

For local explanations, we can see from Figure A4 that, in that particular situation, opposing contributions are made by the cart position (*pos_c*) and velocity (*vel_c*) input features to both reward components. This information may not be obtained if we use the existing method (feature importance and reward decomposition) exclusively. Figure A5 presents the contrastive explanation in which the agent chooses the push-right action instead of push-left. The sum contribution of '$\omega\_p[I]$ to $c\_pos[R]$' and '$\omega\_p[I]$ to $p\_\theta$' is enough to outweigh the disadvantage of not choosing the push-left action. This explanation is further compressed by Algorithm 1 to present that only the contribution of $\omega\_p$ input feature is enough to outweigh the disadvantage of the same situation.

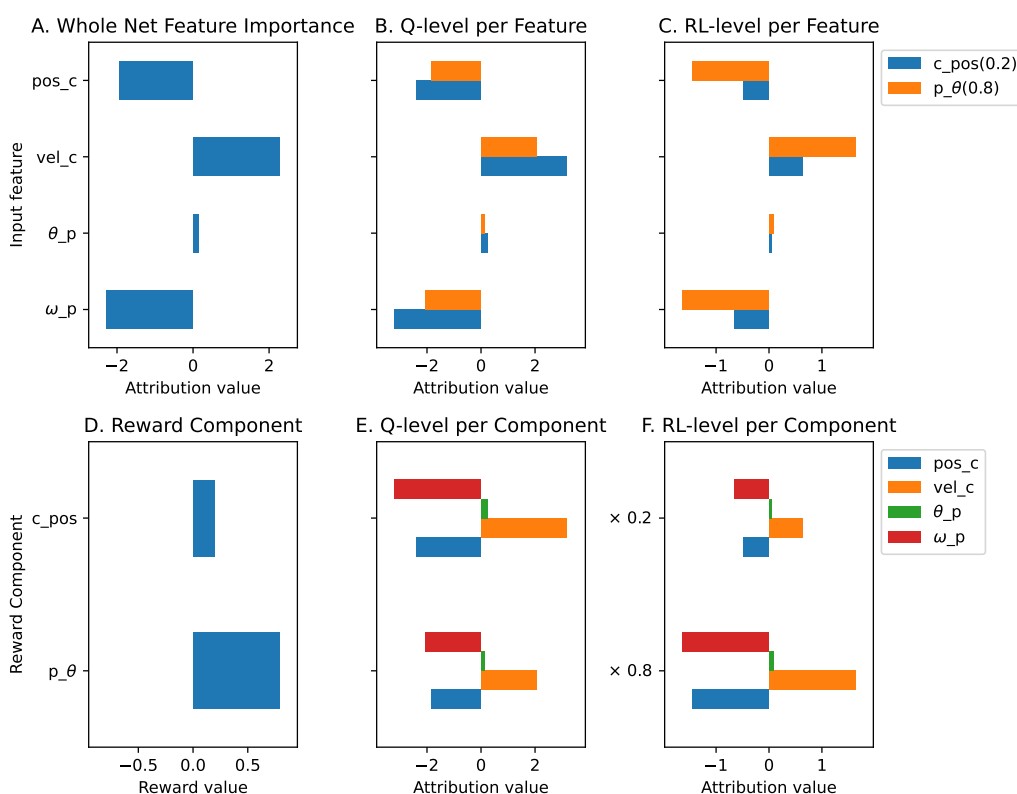

**Figure A4.** The local explanations of the cartpole when the agent chooses the push-right action.

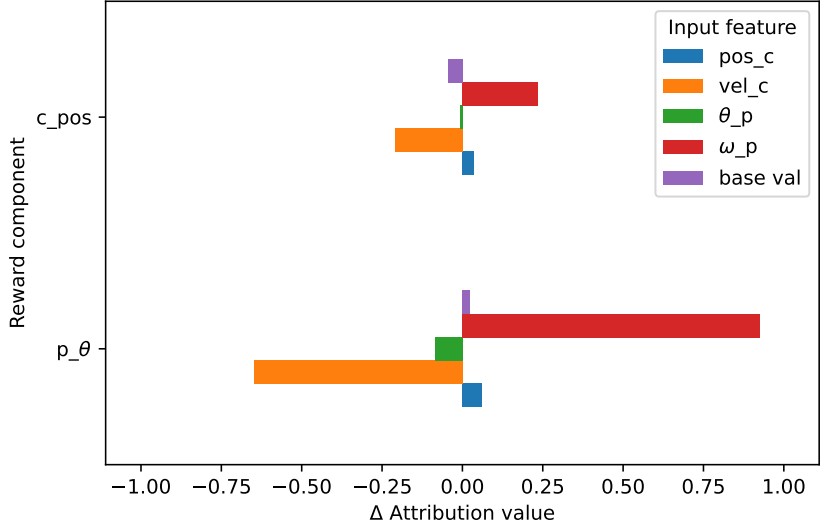

**Figure A5.** The contrastive explanation showing why the agent chooses push-right instead of push-left action.

Table A3 presents the base values of each action and reward component for both in normalized and weighted values. In contrast to the antenna tilt use case, we want this agent to prioritize the angle of the pole more than the position of the cart. Thus, after applying the reward weight, we can see that the pole angle has higher base values than the cart position reward component. We want the cart position and velocity input feature to contribute more to the cart position reward component. Additionally, all input features other than cart position should have a significant impact on the pole angle reward component. These desired properties are set in Table A4 which will be used to generate the focus values as presented in Table A5. We can see that the focus value of the cart position is lower than the

pole angle because there is a significant contribution from the pole velocity input feature to this reward component. The unweighted mean shows the mean of focus values without applying the reward prioritization, while the weighted mean quantifies the fulfillment of the desired properties of the global explanation.

**Table A3.** Base values of the cartpole agent.

| | Unweighted | | Weighted | |
| --- | --- | --- | --- | --- |
| Action | Cart_position | Pole_angle | Cart_position | Pole_angle |
| push_Left | 91.209076 | 85.066811 | 18.241844 | 68.053467 |
| push_right | 90.984283 | 85.097321 | 18.196852 | 68.077972 |
| mean | 91.096680 | 85.082062 | 18.219349 | 68.065720 |
| range | 0.124793 | 0.030510 | 0.044992 | 0.024505 |

**Table A4.** Relevance table for the cartpole agent.

| Reward | Input Feature | | | |
| --- | --- | --- | --- | --- |
| Component | Cart_position | Cart_velocity | Pole_angle | Pole_velocity |
| cart_position | 1.0 | 1.0 | 0.5 | 0.5 |
| pole_angle | 0.5 | 1.0 | 1.0 | 1.0 |

**Table A5.** Focus value of the cartpole agent.

| | Cart_position | Pole_angle | Weighted_mean | Unweighted_mean |
| --- | --- | --- | --- | --- |
| focus value | 0.758693 | 0.944922 | 0.907676 | 0.851807 |

*Appendix D.2. LunarLander from OpenAI Gym*

In this environment, the agent controls the engine to land the lunar lander at the origin $(0,0)$, which is illustrated as a point in the middle of the two flags in Figure A6. Despite being a simple simulator, it formulates a complex problem where more input features are involved and affect differently to different reward components. There are four discrete actions available, which are no operation, fire left, fire up, or fire right. The inputs (state variables) of the RL agent consist of the horizontal coordinate (*pos_x*), vertical coordinate (*pos_y*), horizontal velocity (*vel_x*), vertical velocity (*vel_y*), angle ($\theta$), angular speed ($\omega$), and its legs (left, *c_leg*1 and right, *c_leg*2) contact status, i.e., touching the moon or not.

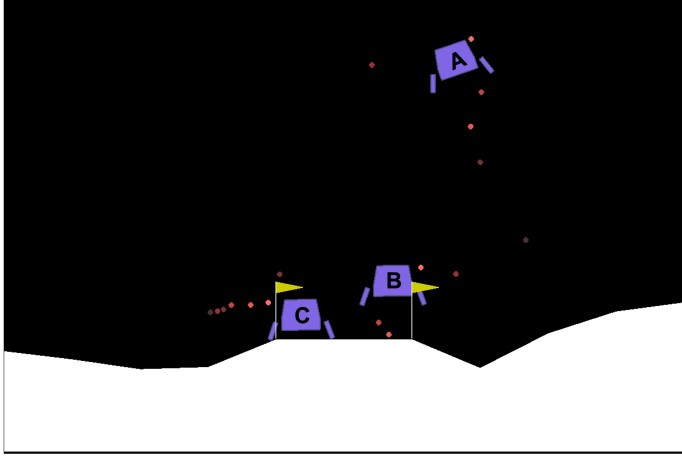

**Figure A6.** The illustration of the Lunarlander environment from openAI gym where the lunar lander (purple) has to land in the middle of yellow flag. A, B, and C are the three states showing the initial, middle, and final phase of the landing process, respectively.

By default, the reward of the lunar lander environment is given as a single value considering the total improvement of all combined factors which are position, velocity, angle, legs contact status, main and side engine activity. To apply our solution, we expose the reward calculation by assigning each factor to the respective reward component ($R_{component}$) function and the reward prioritization. The reward component functions are configured to generate a normalized value, as implemented in the following formulas:

- Position reward: $R_{pos}(s) = -\sqrt{pos\_x^2 + pos\_y^2}$
- Velocity reward: $R_{vel}(s) = -\sqrt{vel\_x^2 + vel\_y^2}$
- Angle reward: $R_{angle}(s) = -|\theta|$

- $Leg_1$ reward: $R_{leg1}(s) = -0.1 \times pos\_y^2 + \begin{cases} 1, & \text{if } c\_leg1 \text{ touch the moon} \\ 0, & \text{otherwise} \end{cases}$

- $Leg_2$ reward: $R_{leg2}(s) = -0.1 \times pos\_y^2 + \begin{cases} 1, & \text{if } c\_leg2 \text{ touch the moon} \\ 0, & \text{otherwise} \end{cases}$

- Main engine reward: $R_{e\_side}(s) = \begin{cases} -1, & \text{if fire up is executed} \\ 0, & \text{otherwise} \end{cases}$

- Side engine reward: $R_{e\_side}(s) = \begin{cases} -1, & \text{if fire left or right is executed} \\ 0, & \text{otherwise} \end{cases}$

For the first five components, we follow the same mechanism as the original implementation to calculate the improvement of each reward component, i.e., $R_{component}(s) = R_{component}(s_t) - R_c(s_{t-1})$ where $t$ denotes the time step. $R_{leg1}(s)$ and $R_{leg2}(s)$ are modified by adding the $pos\_y^2$ calculation ($-0.1 \times pos\_y^2$) while the others remain the same as the original implementation. This is performed to guide the *leg* component to approach the moon surface, which is also performed in [8].

Reward prioritization is implemented as a linear function ($Z_{component} = R_{component}(s) \times k_{component}$) where $k_{component}$ is the weight or multiplication factor of the respective component. In this way, the exact reward weight/prioritization can be presented in a ratio format, i.e., 100 : 100 : 100 : 10 : 10 : 0.30 : 0.03 for $k_{pos} : k_{vel} : k_{angle} : k_{leg1} : k_{leg2} : k_{e\_main} : k_{e\_side}$, respectively. Figure A7 shows two different levels of explanation where we observe that each reward component has input features that contribute to it at the Q-level. However, due to the high difference in reward priority (100 and 0.03), the contributions of input features to the main and side engine reward components are barely visible at the RL-level. Without decoupling the reward priority from the reward function, the NN of the Q-function can generate a high value (up to 2000), which is susceptible to gradient explosion. In contrast, when the NN output is too low, it is susceptible to a vanishing gradient problem.

Figure A7B shows the detailed explanations on how each input feature contributes to every reward component. Input feature $pos\_y$ contributes significantly to the position and the main engine reward components, which is desirable. We can also see that both the leg-contact input features ($c\_leg1$ and $c\_leg2$) contribute significantly to the respective leg reward components. Figure A8 shows the comparison of the input feature and the reward component at two different levels. We can see that when reward prioritization is not applied (Figure A8a), several elements have significant contributions in generating the Q-values prediction. On the other hand, when reward prioritization is applied (Figure A8b), the input feature $pos\_y$ and the position reward component contribute the highest compared to others.

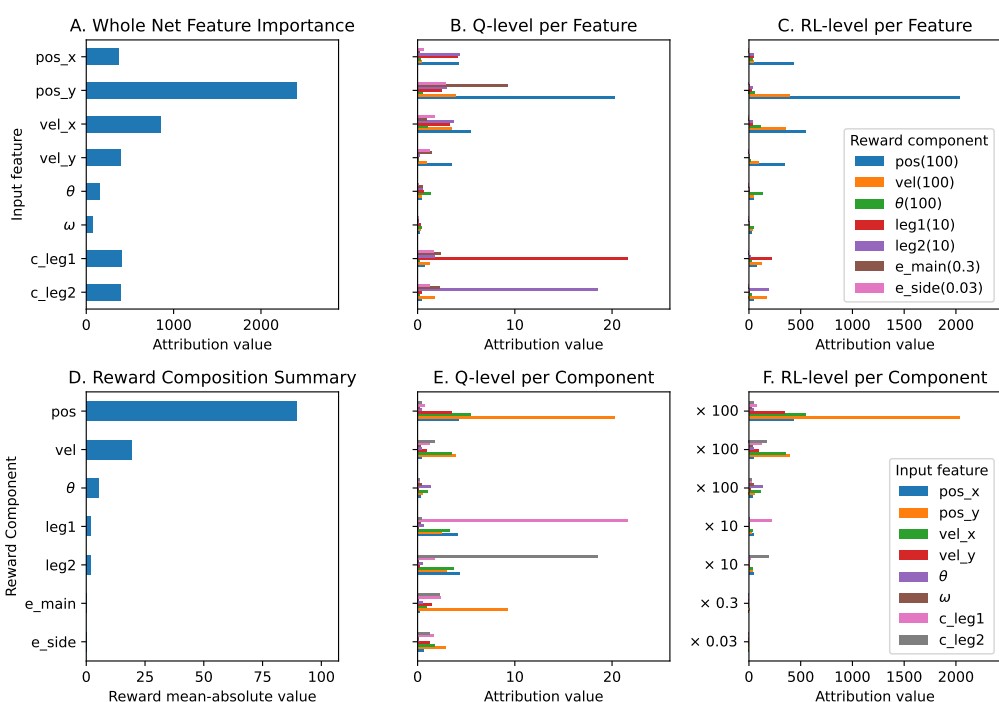

**Figure A7.** The global explanations of the Lunarlander agent.

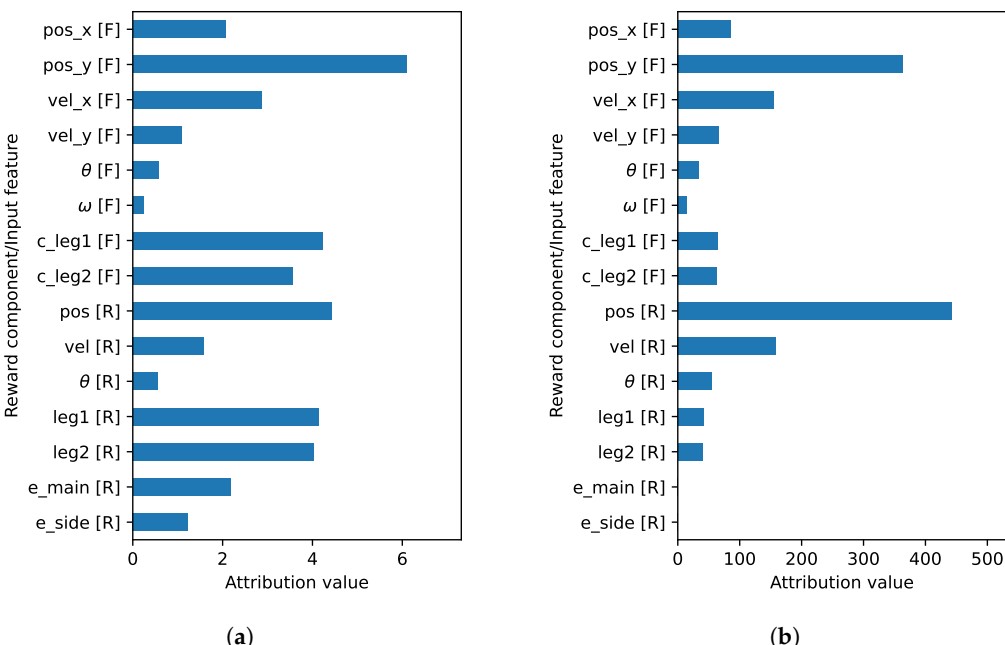

(**a**)                                                                 (**b**)

**Figure A8.** The normalized explanations of the Lunarlander agent where the importance of the input features and the reward components are compared in the same plot. (**a**) Normalized explanation at the Q-function level. (**b**) Normalized explanation when the reward weight is applied (at RL-level).

In a local scope, Figure A9 shows the explanations of two different states of the lunar lander. Figure A9a show the explanations for choosing the fire_right action in the initial phase of the landing process. We can see that pos_y input feature contributes negatively to several reward components and at the RL level it significantly contributes to the position and velocity reward component. This explanation makes sense because the lander is still far from the moon's surface. Furthermore, we can also see that the angle ($\theta$) input feature contributes negatively to the angle ($\theta$) reward component because the lander is rotated. The explanation of when the lander has landed on the moon (Figure A6 position C) is

shown in Figure A9b. At the Q-level explanation, we can see that the contributions of the leg contact input features that contribute to the respective leg reward components are significant compared to others. However, at the RL-level, the contribution of pos_y input feature to the position reward component is the highest due to reward prioritization, while the contributions of the respective legs elements become the second and third.

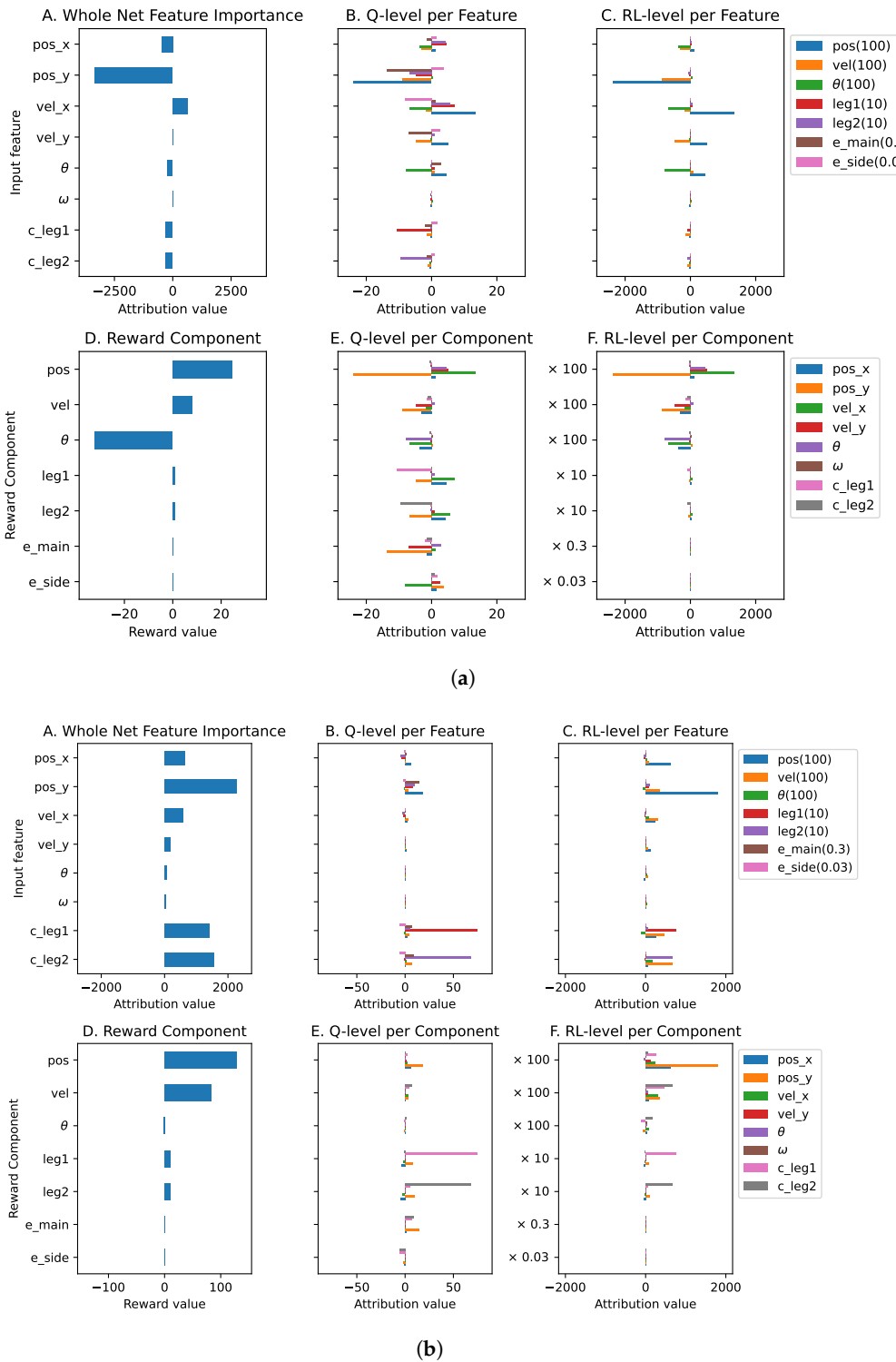

**Figure A9.** The local explanations of the Lunarlander of three different states as shown in Figure A6. (**a**) The local explanations of the Lunarlander when the agent chooses fire_right action on position A in Figure A6. (**b**) The local explanations of the Lunarlander when the agent chooses fire_left action on position C in Figure A6.

When comparing two actions, the contrastive explanation shows the benefit of choosing an action over the other, as shown in Figure A10. By applying the MSX concept, the contributions of *"pos_x[I] to pos[R]"*, *"pos_x[I] to vel[R]"*, *"vel_y[I] to vel[R]"*, and *"c_leg1[I] to vel[R]"* are enough to choose fire_right action and outweigh the disadvantage of not choosing no_fire action. Furthermore, our automatic MSX compression (Algorithm 1) reduces the length of the explanation to only present the contributions of position and velocity reward components to be presented to the user.

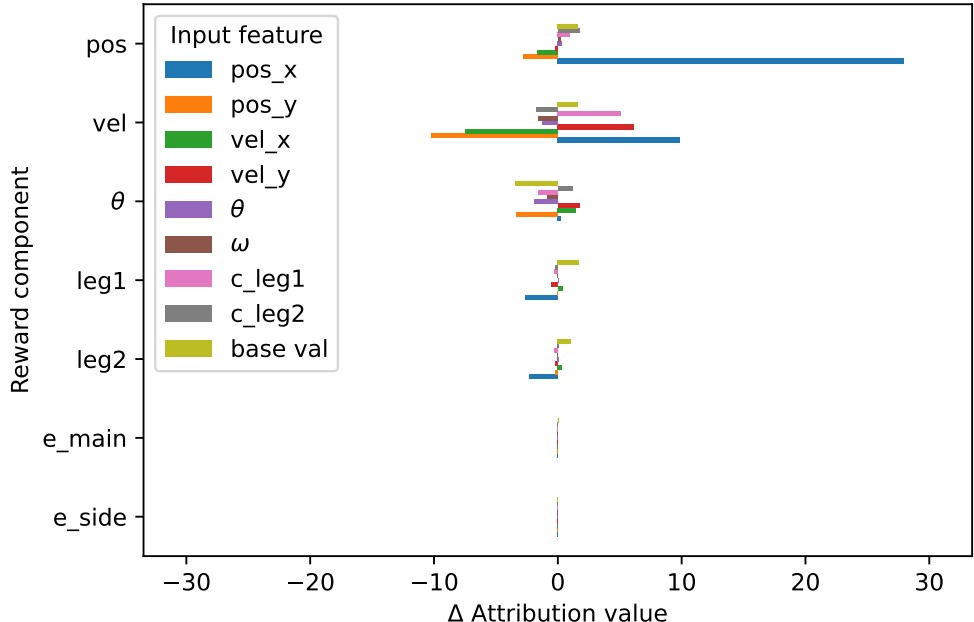

**Figure A10.** The contrastive explanation showing why the agent chooses fire_right instead of no_fire action on position B in Figure A6.

The base value of each pair of action and reward components is shown in Tables A6 and A7, where the former is calculated at the Q-level while the latter is at the RL-level. We can see that the range at the RL-level is significantly larger than the Q-level because of the reward prioritization. The base value of the position reward component has the highest value, which is relevant to this environment where the main goal is to position the lander to the origin.

**Table A6.** Base values of the Lunarlander agent without reward prioritization.

| Reward | Action | | | |
|---|---|---|---|---|
| Component | No_fire | Fire_right | Fire_up | Fire_left |
| position | 99.954918 | 99.971298 | 99.855209 | 100.127533 |
| velocity | 24.920115 | 24.936441 | 25.133207 | 24.831766 |
| angle | −3.527749 | −3.562116 | −3.443951 | −3.553954 |
| leg1 | 23.622988 | 23.793665 | 23.522532 | 23.526903 |
| leg2 | 22.642960 | 22.748714 | 22.536240 | 22.465919 |
| main_engine | −36.977737 | −36.853558 | −37.329254 | −36.843079 |
| side_engine | −50.239410 | −51.475277 | −50.376621 | −51.710163 |

All above explanations are detailed and the focus value can be used to summarize whether the explanations fulfill the desired properties. For this environment, the relevance table is set as shown in Table A8 which generates the focus value shown in Table A9. The position component ($R_{pos}$) is highly correlated with the position input features ($pos\_x$, $pos\_y$). Similarly, other pairs are also highly correlated where we set the relevance value as 1

such as: velocity component ($R_{vel}$) with speed ($speed_h, speed_v, speed_a$); angle component ($R_{angle}$) with angular speed ($speed_a$) and angle input feature; and leg components ($R_{leg1}$ and $R_{leg2}$) with leg input feature ($leg_1$ and $leg_2$, respectively), etc. On the other hand, the main engine ($R_{main-eng}$) should not consider the angular speed ($speed_a$) because changes in the angular speed cannot be corrected by the action of the main engine; therefore, it has zero relevance value.

**Table A7.** Base values of the Lunarlander agent when reward prioritization is applied.

| Reward | Action | | | |
|---|---|---|---|---|
| Component | No_fire | Fire_right | Fire_up | Fire_left |
| position | 9995.485352 | 9997.129883 | 9985.517578 | 10012.757812 |
| velocity | 2492.011719 | 2493.641357 | 2513.325684 | 2483.169189 |
| angle | −352.775330 | −356.211731 | −344.395172 | −355.395264 |
| leg1 | 236.229767 | 237.936707 | 235.225174 | 235.268982 |
| leg2 | 226.429245 | 227.487045 | 225.362274 | 224.659210 |
| main_engine | −11.093332 | −11.056075 | −11.198791 | −11.052940 |
| side_engine | −1.507182 | −1.544255 | −1.511300 | −1.551306 |

**Table A8.** Relevance table for the Lunarlander agent.

| Reward | Input Feature | | | | | | | |
|---|---|---|---|---|---|---|---|---|
| Comp. | Pos_x | Pos_y | Vel_x | Vel_y | $\theta$ | $\omega$ | c_leg1 | c_leg2 |
| position | 1.0 | 1.0 | 0.5 | 0.5 | 0.5 | 0.5 | 0.5 | 0.5 |
| velocity | 0.5 | 0.5 | 1.0 | 1.0 | 0.5 | 1.0 | 0.5 | 0.5 |
| angle($\theta$) | 0.5 | 0.5 | 0.5 | 0.5 | 1.0 | 1.0 | 0.5 | 0.5 |
| leg1 | 0.5 | 0.5 | 0.5 | 0.5 | 0.5 | 0.5 | 1.0 | 0.5 |
| leg2 | 0.5 | 0.5 | 0.5 | 0.5 | 0.5 | 0.5 | 0.5 | 1.0 |
| main_eng | 0.5 | 1.0 | 0.5 | 1.0 | 0.5 | 0.0 | 0.5 | 0.5 |
| side_eng | 1.0 | 0.5 | 1.0 | 0.5 | 1.0 | 1.0 | 1.0 | 1.0 |

**Table A9.** Focus value of the Lunarlander agent.

| Reward Component | Focus Value |
|---|---|
| position | 0.847708 |
| velocity | 0.690510 |
| angle | 0.698298 |
| leg1 | 0.826667 |
| leg2 | 0.787410 |
| main_engine | 0.804658 |
| side_engine | 0.784765 |
| weighted_mean | 0.749406 |
| unweighted_mean | 0.777145 |

We can see that the position reward component has the highest focus value, and *pos_y* contributes significantly higher than the others. The velocity ($R_{vel}$) reward component has a low value because *pos_y* has a higher contribution than *vel_x*, *vel_y* or the angular velocity ($\omega$), which ideally is the focus of this component. Similarly, the angle reward component has a low value because *vel_x* and *pos_y* are the second and third most contributing feature instead of the angular velocity ($\omega$). Overall, the weighted mean has a lower value than the unweighted mean focus value, which means that high priority components (i.e., velocity and angle) do not focus on the desired input features.

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
