# Peer review of "BEERL: Both Ends Explanations for Reinforcement Learning"

_applsci, doi:10.3390/app122110947_

Round 1

Reviewer 1 Report

The significance of content of the article can be improved by some additional practical examples.

Author Response

Authors' Response: Previously, the implementation of this method was done to two use cases: 1. Remote electric tilt use case from telecom domain, presented in Section 4, and 2. cartpole example from open AI gym (presented in the appendix).

We have now added another example (lunar lander environment) in the end of the appendix to show that our method can also be applied to other use cases.

Reviewer 2 Report

The work exerts XAI approaches to correlate input and output explanations for agents that apply RL. It exposes the reward prioritization to identify two levels of explanation and allows the reconfiguration of RL agents. The authors also present focus-value weighting properties and their contribution to the output behavior. Finally, they apply the method and show the results of the remote electrical antenna tilt use case.

The result section is good. However, this reviewer's main concern is that the paper is hard to follow. In addition, extensive editing of English language and style is required. Some sections need to be reorganized. Literature review and related work need to be better explained.

When the authors mentioned in the abstract that "Explainable artificial intelligence (XAI) is a technique that tries to explain the behavior and decisions made by an artificial intelligence (AI) agent" -->  this statement is not correct, XAI is general to AI and not specific to agents.

Also "Deep Reinforcement Learning is such a black-box method, especially when a Reinforcement Learning (RL) agent employs a neural network (NN)" --> RL always used NN

"resulting in a finer granularity of the explanation." --> granularity seems odd not related to the context.

In the introduction, the following contribution is not clear "3) extend the contrastive explanation method to simplify the explanations while fulfilling the minimal sufficient explanation MSX concept."

Section 2.2.1 "for the explainer for our method to explain the input side of the model. However, instead of directly feeding the RL model's output to the explainer, our method feeds the second-last layer to generate the explanations" --> this sentence needs to be rewritten and clarified. It is very confusing. Which second layer? Shouldn't this be the explainability of the input features as the subtitle of the section indicate?

Section 3 page 4 "And to achieve this goal …." don't start a sentence with an "And"

Section 3 flow is messy and confusing. When the authors explain in the user config. Section 3.2 that the user receives 4 types: 1) Q function is section 3.3.1, 2) reward prioritization is section 3.2.1, 3) RL explanation - which subsection does this belong to?, 4) focus values ?

The reader is expecting the subsections to follow the 4 types however these 4 types are provided in a scattered manner and in other subsections which is very confusing, breaking the flow and very hard to follow.

Lines 213-221 is too wordy and can be compacted

Section 3.3. RL Agent "are shown in Figure 1" --> which part of the figure?

The paragraph at Line 293 -- is unclear

The equations 5 & 6 explain what is C, I, T etc…

Algorithm 1 is not explained in the tex

Show the units in the figures. Example figure 4.

Reviewer 3 Report

1.       No detailed citation is required in the abstract. The problems mainly focus on lines 4-8.

2.       It is suggested to supplement key quantitative data in the abstract to intuitively reflect the progressiveness of the method, which will help readers to quickly understand the performance of the method.

3.       Is the concept of the component bis (line 41) proposed by the author? If not, add a reference.

4.       Please delete some sentences with subjective meaning, such as “to the best of the authors' knowledge” (96 lines). Please check the full text carefully and avoid similar problems.

5.       Don't explain the existing concepts too much in the methodology. This is not a review paper! The author should introduce the proposed framework and elements directly. This will save a lot of space.

6.       In Section 3.3.1, why can the proposed method has the benefit of avoiding vanishing/exploding gradients for the neural network? Please provide reasonable explanation and experimental proof.

7.       Why the relevant work is discussed again in Section 4.1

8.       Please adjust the size of the text in the picture for the convenience of readers. The problem occurs in almost all the graphs that reflect the data comparison.

9.       In some figures in the appendix (e.g. Fig. A2-A4), the legend obscures the data. Please check carefully and correct.

Round 2

Reviewer 2 Report

I would like to thank the authors for addressing all the issues and improving the paper quality

Reviewer 3 Report

I would like to thank the authors for their very comprehensive and convincing responses.